# Flexible learning of quantum states with generative query neural networks

Yan Zhu [1,6], Ya-Dong Wu [1,6] ✉, Ge Bai [1], Dong-Sheng Wang[2], Yuexuan Wang[1,3] & Giulio Chiribella [1,4,5] ✉

Deep neural networks are a powerful tool for characterizing quantum states. Existing networks are typically trained with experimental data gathered from the quantum state that needs to be characterized. But is it possible to train a neural network offline, on a different set of states? Here we introduce a network that can be trained with classically simulated data from a fiducial set of states and measurements, and can later be used to characterize quantum states that share structural similarities with the fiducial states. With little guidance of quantum physics, the network builds its own data-driven representation of a quantum state, and then uses it to predict the outcome statistics of quantum measurements that have not been performed yet. The state representations produced by the network can also be used for tasks beyond the prediction of outcome statistics, including clustering of quantum states and identification of different phases of matter.

Accurate characterization of quantum hardware is crucial for the development, certification, and benchmarking of new quantum technologies[1]. Accordingly, major efforts have been invested into developing suitable techniques for characterizing quantum states, including quantum state tomography[2-6], classical shadow estimation[7,8], partial state characterization[9,10], and quantum state learning[11-14]. Recently, the dramatic development of artificial intelligence inspired new approaches on machine learning methods[15]. In particular, a sequence of works explored applications of neural networks to various state characterization tasks[16-26].

In the existing quantum applications, neural networks are typically trained using experimental data generated from the specific quantum state that needs to be characterized. As a consequence, the information learnt in the training phase cannot be directly transferred to other states: for a new quantum state, a new training procedure must be carried out. This structural limitation affects the learning efficiency in applications involving multiple quantum states, including important tasks such as quantum state clustering[27], quantum state classification[28], and quantum cross-platform verification[29].

In this paper, we develop a flexible model of neural network that can be trained offline using simulated data from a fiducial set of states and measurements, and is capable of learning multiple quantum states that share structural similarities with the fiducial states, such as being ground states in the same phase of a quantum manybody system.

## Results

### Quantum state learning framework

In this work we adopt a learning framework inspired by the task of "pretty good tomography"[11]. An experimenter has a source that produces quantum systems in some unknown quantum state $\rho$. The experimenter's goal is to characterize $\rho$, becoming able to make predictions on the outcome statistics of a set of measurements of interest, denoted by $\mathcal{M}$. Each measurement $\boldsymbol{M} \in \mathcal{M}$ corresponds to a positive operator-valued measure (POVM), that is, a set of positive operators $\boldsymbol{M} := (M_j)_{j=1}^{n_o}$ acting on the system's Hilbert space and satisfying the normalization condition $\sum_{j=1}^{n_o} M_j = \mathbb{1}$ (without loss of generality, we assume that all the measurements in $\mathcal{M}$ have the same number of outcomes, denoted by $n_o$).

[1]QICI Quantum Information and Computation Initiative, Department of Computer Science, The University of Hong Kong, Pokfulam, Hong Kong. [2]CAS Key Laboratory of Theoretical Physics, Institute of Theoretical Physics, Chinese Academy of Sciences, Beijing 100190, P.R. China. [3]College of Computer Science and Technology, Zhejiang University, Hangzhou, China. [4]Department of Computer Science, Oxford OX1 3QD, UK. [5]Perimeter Institute for Theoretical Physics, Waterloo, ON N2L 2Y5, Canada. [6]These authors contributed equally: Yan Zhu, Ya-Dong Wu. ✉e-mail: yadongwu@hku.hk; giulio@cs.hku.hk

To characterize the state $\rho$, the experimenter performs a finite number of measurements $\boldsymbol{M}_i$, $i \in \{1, ..., s\}$, picked at random from $\mathcal{M}$. This random subset of measurements will be denoted by $\mathcal{S} = \{\boldsymbol{M}_i\}_{i=1}^s$. Note that in general both $\mathcal{M}$ and $\mathcal{S}$ may not be informationally complete.

Each measurement in $\mathcal{S}$ is performed multiple times on independent copies of the quantum state $\rho$, obtaining a vector of experimental frequencies $\mathbf{p}_i$. Using this data, the experimenter attempts to predict the outcome statistics of a new, randomly chosen measurement $\boldsymbol{M}' \in \mathcal{M} \setminus \mathcal{S}$. For this purpose, the experimenter uses the assistance of an automated learning system (e.g. a neural network), hereafter called the learner. For each measurement $\boldsymbol{M}_i \in \mathcal{S}$, the experimenter provides the learner with a pair $(\mathbf{m}_i, \mathbf{p}_i)$, where $\mathbf{m}_i$ is a parametrization of the measurement $\boldsymbol{M}_i$, and $\mathbf{p}_i$ is the vector of experimental frequencies for the measurement $\boldsymbol{M}_i$. Here the parametrization $\mathbf{m}_i$ could be the full description of the POVM $\boldsymbol{M}_i$, or a lower-dimensional parametrization valid only for measurements in the set $\mathcal{M}$. For example, if $\mathcal{M}$ contains measurements of linear polarization, a measurement in $\mathcal{M}$ could be parametrized by the angle $\theta$ of the corresponding polarizer. The parametrization could also be encrypted, so that the actual description of the quantum hardware in the experimenter's laboratory is concealed from the learner. In the following, we denote by enc the function mapping a POVM $\boldsymbol{M} \in \mathcal{M}$ into its parametrization enc($\boldsymbol{M}$). With this notation, we have $\mathbf{m}_i = \text{enc}(\boldsymbol{M}_i)$ for every $i \in \{1, ..., s\}$.

To obtain a prediction for a new, randomly chosen measurement $\boldsymbol{M}' \in \mathcal{M} \setminus \mathcal{S}$, the experimenter sends its parametrization $\mathbf{m}' := \text{enc}(\boldsymbol{M}')$ to the learner. The learner's task is to predict the correct outcome probabilities $\mathbf{p}'_{\text{true}} = (\text{tr}(\rho M'_j))_{j=1}^{n_o}$. This task includes as special case quantum state reconstruction, corresponding to the situation where the subset $\mathcal{S}$ is informationally complete.

Note that, a priori, the learner may have no knowledge about quantum physics whatsoever. The ability to make reliable predictions about the statistics of quantum measurements can be gained automatically through a training phase, where the learner is presented with data and adjusts its internal parameters in a data-driven way. In previous works[16,17,19,20,24,26], the training was based on experimental data gathered from the same state $\rho$ that needs to be characterized. In the following, we will provide a model of learner that can be trained with data from a fiducial set of quantum states that share some common structure with $\rho$, but can generally be different from $\rho$. The density matrices of the fiducial states can be completely unknown to the learner. In fact, the learner does not even need to be provided a parametrization of the fiducial states: the only piece of information that the learner needs to know is which measurement data correspond to the same state.

## The GQNQ network

Our model of learner, GQNQ, is a neural network composed of two main parts: a representation network[30], producing a data-driven representation of quantum states, and a generation network[31], making predictions about the outcome probabilities of quantum measurements that have not been performed yet. The combination of a representation network and a generation network is called a generative query network[32]. This type of neural network was originally developed for the classical task of learning 3D scenes from 2D snapshots taken from different viewpoints. The intuition for adapting this model to the quantum domain is that the statistics of a fixed quantum measurement can be regarded as a lower-dimensional projection of a higher-dimensional object (the quantum state), in a way that is analogous to a 2D projection of a 3D scene. The numerical experiments reported in this paper indicate that this intuition is indeed correct, and that GQNQ works well even in the presence of errors in the measurement data and fluctuations due to finite statistics.

The structure of GQNQ is illustrated in Fig. 1. The first step is to produce a representation $\mathbf{r}$ of the unknown quantum state $\rho$. In GQNQ,

this step is carried out by a representation network, which computes a function $f_{\boldsymbol{\xi}}$ depending on parameters $\boldsymbol{\xi}$ that are fixed after the training phase (see Methods for details). The representation network receives as input the parametrization of all measurements in $\mathcal{S}$ and their outcome statistics on the specific state $\rho$ that needs to be characterized. For each pair $(\mathbf{m}_i, \mathbf{p}_i)$, the representation network produces a vector $\mathbf{r}_i = f_{\boldsymbol{\xi}}(\mathbf{m}_i, \mathbf{p}_i)$. The vectors corresponding to different pairs are then combined into a single vector $\mathbf{r}$ by an aggregate function $\mathcal{A}$. Here, we take the aggregate function to be the average, namely $\mathbf{r} := \frac{1}{s} \sum_{i=1}^s \mathbf{r}_i$.

When GQNQ is used to characterize multiple quantum states $\rho^{(j)}$, $j \in \{1, ..., K\}$, the above procedure is repeated for each state $\rho^{(j)}$. To characterize the state $\rho^{(j)}$, GQNQ uses measurement data generated from a subset of $s$ measurements $\mathcal{S}^{(j)} \subset \mathcal{M}$, which in general we allow to depend on $j$. For the the $i$-th measurement in $\mathcal{S}^{(j)}$, denoted by $\boldsymbol{M}_i^{(j)}$, GQNQ produces a representation vector $\mathbf{r}_i^{(j)} := f_{\boldsymbol{\xi}}(\mathbf{m}_i^{(j)}, \mathbf{p}_i^{(j)})$, where $\mathbf{m}_i^{(j)} := \text{enc}(\boldsymbol{M}_i^{(j)})$ is the parametrization of the measurement $\boldsymbol{M}_i^{(j)}$ and $\mathbf{p}_i^{(j)}$ is the measurement statistics obtained by performing $\boldsymbol{M}_i^{(j)}$ on the state $\rho^{(j)}$. The representation vectors $\mathbf{r}_i^{(j)}$ are then combined into a a single vector $\mathbf{r}^{(j)} := \sum_{i=1}^s \mathbf{r}_i^{(j)}/s$, which serves as a representation of the state $\rho^{(j)}$. Note that the vectors $\mathbf{r}_i^{(j)}$, and therefore the state representation $\mathbf{r}^{(j)}$, depend only on the outcome statistics of measurements performed on the state $\rho^{(j)}$, and on parameters $\boldsymbol{\xi}$ that are fixed after the training phase. As a consequence, the state representation $\mathbf{r}^{(j)}$ does not depend on measurement data associated to states $\rho^{(l)}$ with $l \neq j$.

Once a state representation has been produced, the next step is to predict the outcome statistics for a new quantum measurement on that state. In quantum tomography, the prediction is generated by applying the Born rule on the estimated density matrix. In GQNQ, the task is achieved by a generation network[32], which computes a function $g_{\boldsymbol{\eta}}$ depending on some parameters $\boldsymbol{\eta}$ that are fixed after the training phase. To make predictions about the state $\rho^{(j)}$, the network receives as input the state representation $\mathbf{r}^{(j)}$ and the parametrization $\mathbf{m}'$ of the desired measurement $\boldsymbol{M}' \in \mathcal{M} \setminus \mathcal{S}^{(j)}$. The output of the generation network is a vector $\mathbf{p}' := g_{\boldsymbol{\eta}}(\mathbf{r}^{(j)}, \mathbf{m}')$ that approximates the outcome statistics of the measurement $\boldsymbol{M}'$ on the state $\rho^{(j)}$.

Crucially, GQNQ does not need any parametrization of the quantum states $(\rho^{(j)})_{j=1}^K$, neither it needs the states to be sorted into different classes. For example, if the states correspond to different phases of matter, GQNQ does not need to be told which state belongs to which phase. This feature will be important for the applications to state clustering and classification illustrated later in this paper.

The exact form of the functions $f_{\boldsymbol{\xi}}$ and $g_{\boldsymbol{\eta}}$ is determined by the internal structure of the representation and generation networks, provided in Supplementary Note 1. The purpose of the training phase is to choose appropriate values of the parameters $\boldsymbol{\xi}$ and $\boldsymbol{\eta}$. In the training phase, GQNQ starts from some randomly chosen initial values $\boldsymbol{\xi} = \boldsymbol{\xi}_0$ and $\boldsymbol{\eta} = \boldsymbol{\eta}_0$, and then updates the values of $\boldsymbol{\xi}$ and $\boldsymbol{\eta}$ through a gradient descent procedure (see Methods for details). To implement this procedure, GQNQ is provided with pairs $(\mathbf{m}, \mathbf{p})$ consisting of the measurement parametrization/measurement statistics for a fiducial set of measurements $\mathcal{M}_* \subseteq \mathcal{M}$, performed on a fiducial set of quantum states $\mathcal{Q}_*$. In the numerical experiments provided in the Results section, we choose $\mathcal{M}_* = \mathcal{M}$, that is, we provide the network with the statistics of all the measurement in $\mathcal{M}$. In the typical scenario, the fiducial states and measurements are known, and the training can be done offline, using computer simulated data rather than actual experimental data.

We stress that the parameters $\boldsymbol{\xi}$ and $\boldsymbol{\eta}$ depend only on the fiducial sets $\mathcal{M}_*$ and $\mathcal{Q}_*$ and on the corresponding measurement data, but do not depend on the unknown quantum states $(\rho^{(j)})_{j=1}^K$ that will be characterized later, nor on the subsets of measurements $(\mathcal{S}^{(j)})_{j=1}^K$ that will be performed on these states. Hence, the network does not need to be re-trained when it is used to characterize a new quantum state $\rho^{(j)}$, nor to be re-trained when one changes the subset of performed measurements $\mathcal{S}^{(j)}$.

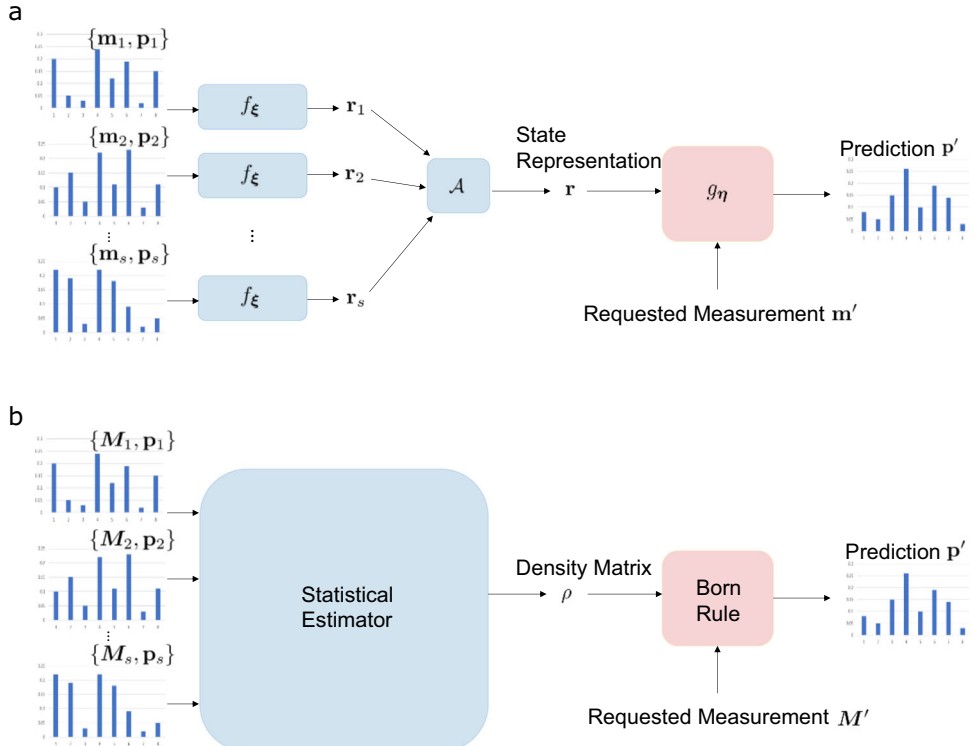

**Fig. 1 | Structure of GQNQ and comparison with quantum state tomography.** In GQNQ (**a**), a representation network receives as input the raw measurement data $\{(\mathbf{m}_i, \mathbf{p}_i)\}_{i=1}^{s}$ and produces as output $s$ vectors $\mathbf{r}_i = f_{\xi}(\mathbf{m}_i, \mathbf{p}_i)$, that are combined into a single vector $\mathbf{r}$ by an aggregate function $\mathcal{A}$. The vector $\mathbf{r}$ serves as a concise representation of the quantum state, and is sent to a generation network $g_{\boldsymbol{\eta}}$, which predicts the outcome statistics $\mathbf{p}'$ of any desired measurement $\mathbf{m}'$ in the set of measurements of interest. In quantum tomography (**b**), the raw measurement data are fed into a statistical estimator (such as maximum likelihood), which produces a guess for the density matrix $\rho$. Then, the density matrix is used to predict the outcome probabilities of unperformed quantum measurements via the Born rule. Both GQNQ and quantum tomography use data to infer a representation of the quantum state.

It is useful to contrast GQNQ with quantum tomography. While tomographic protocols strive to find the density matrix that fits the measurement data, GQNQ is not constrained to a specific choice of state representation. This additional freedom enables the network to construct lower-dimensional representations of quantum states with sufficiently regular structure, such as ground states in well-defined phases of matter, and to make predictions for states that did not appear in the training phase. Notice also that the tomographic reconstruction of the density matrix using statistical estimators, such as maximum likelihood and maximum entropy[33], is generally more time-consuming than the evaluation of the function $f_{\xi}$, due to the computational complexity of the estimation procedure.

Another difference with quantum tomography is that GQNQ does not require a specific representation of quantum measurements in terms of POVM operators. Instead, a measurement parametrization is sufficient for GQNQ to make its predictions, and the parametrization can even be provided in an encrypted form. Since GQNQ does not require the description of the devices to be provided in clear, it can be used to perform data analysis on a public server, without revealing properties of the quantum hardware, such as the dimension of the underlying quantum system.

Summarizing, the main structural features of GQNQ are
- Offline, multi-purpose training: training can be done offline using computer generated data. Once the training has been concluded, the network can be used to characterize and compare multiple states.
- Measurement flexibility: after the training has been completed, the experimenter can freely choose which subset of measurements $\mathcal{S} \subset \mathcal{M}$ is performed on the unknown quantum states.

- Learner-blindness: the parametrization of the measurements can be provided in an encrypted form. No parametrization of the states is needed.

Later in the paper, we will show that GQNQ can be adapted to an online version of the state learning task[13], thus achieving the additional feature of
- Online prediction: predictions can be updated as new measurement data become available.

## Quantum state learning in spin systems

A natural test bed for our neural network model is provided by quantum spin systems[34,35]. In the following, we consider ground states of the one-dimensional transverse-field Ising model and of the XXZ model, both of which are significant for many-body quantum simulations[36–38]. These two models correspond to the Hamiltonians

$$H = -\left(\sum_{i=0}^{L-2} J_i \sigma_i^z \sigma_{i+1}^z + \sum_{j=0}^{L-1} \sigma_j^x\right), \tag{1}$$

and

$$H = -\left[\sum_{i=0}^{L-2} \Delta_i \left(\sigma_i^x \sigma_{i+1}^x + \sigma_i^y \sigma_{i+1}^y\right) + \sigma_i^z \sigma_{i+1}^z\right], \tag{2}$$

respectively. In the Ising Hamiltonian (1), positive (negative) coupling parameters $J_i$ correspond to ferromagnetic (antiferromagnetic) interactions. For the XXZ Hamiltonian (2), the ferromagnetic phase corresponds to coupling parameters $\Delta_i$ in the interval $(-1, 1)$. If instead the

**Table 1 | Average classical fidelities between the predictions of GQNQs and the ground truths with respect to different types of six-qubit states**

| Types of states for training and test | Noiseless | 50 shots | 10 shots |
|---|---|---|---|
| (i) Ising ground states with ferromagnetic bias | 0.9870 | 0.9869 | 0.9862 |
| (ii) Ising ground states with antiferromagnetic bias | 0.9869 | 0.9867 | 0.9849 |
| (iii) Ising ground states with no bias | 0.9895 | 0.9894 | 0.9894 |
| (iv) XXZ ground states with ferromagnetic bias | 0.9809 | 0.9802 | 0.9787 |
| (v) XXZ ground states with XY phase bias | 0.9601 | 0.9548 | 0.9516 |
| (vi) (i)-(v) together | 0.9567 | 0.9547 | 0.9429 |
| (vii) GHZ state with local rotations | 0.9744 | 0.9744 | 0.9742 |
| (viii) W state with local rotations | 0.9828 | 0.9826 | 0.9821 |
| (ix) (i)–(v), (vii) and (viii) together | 0.9561 | 0.9543 | 0.9402 |

coupling parameters fall in the region $(-\infty, -1) \cup (1, \infty)$, the Hamiltonian is said to be in the XY phase[39].

We start by considering a system of six qubits as example. For the ground states of the Ising model (1), we choose each coupling parameter $J_i$ at random following a Gaussian distribution with standard deviation $\sigma = 0.1$ and mean $J$. For $J > 0$ ($J < 0$), this random procedure has a bias towards ferromagnetic (antiferromagnetic) interactions. For $J = 0$, ferromagnetic and antiferromagnetic interactions are equally likely. Similarly, for the ground states of the XXZ model (2), we choose each parameter $\Delta_i$ at random following a Gaussian distribution with standard deviation 0.1 and mean value $\Delta$. When $\Delta$ is in the interval $(-1, 1)$ $((-\infty, -1) \cup (1, \infty))$, this random procedure has a bias towards interactions of the ferromagnetic (XY) type.

In addition to the above ground states, we also consider locally rotated GHZ states, of the form $\otimes_{i=1}^{6} U_i |\mathrm{GHZ}\rangle$ with $|\mathrm{GHZ}\rangle = \frac{1}{\sqrt{2}}(|000000\rangle + |111111\rangle)$ and locally rotated W states, of the form $\otimes_{i=1}^{6} U_i |\mathrm{W}\rangle$ with $|\mathrm{W}\rangle = \frac{1}{\sqrt{6}}(|100000\rangle + \cdots + |000001\rangle)$, where $(U_i)_{i=1}^{6}$ are unitary matrices of the form $U_i = \exp[-i\theta_{i,z}\sigma_{i,z}] \exp[-i\theta_{i,y}\sigma_{i,y}] \exp[-i\theta_{i,x}\sigma_{i,x}]$, where the angles $\theta_{i,x}, \theta_{i,y}, \theta_{i,z} \in [0, \pi/10]$ are chosen independently and uniformly at random for every $i$.

For the set of all possible measurements $\mathcal{M}$, we chose the 729 six-qubit measurements consisting of local Pauli measurements on each qubit. To parameterize the measurements in $\mathcal{M}$, we provide the entries in the corresponding Pauli matrix at each qubit, arranging the entries in a 48-dimensional real vector. The dimension of state representation **r** is set to be 32, which is half of the Hilbert space dimension. In Supplementary Note 2 we discuss how the choice of dimension of **r** and the other parameters of the network affect the performance of GQNQ.

GQNQ is trained using measurement data from measurements in $\mathcal{M}$ on states of the above four types (see Methods for a discussion of the data generation techniques). We consider both the scenarios where all training data come from states of the same type, and where states of different types are used. In the latter case, we do not provide the network with any label of the state type. After training, we test GQNQ on states of the four types described above. To evaluate the performance of the network, we compute the classical fidelities between the predicted probability distributions and the correct distributions computed from the true states and measurements. For each test state, the classical fidelity is averaged over all possible measurements in $\mathcal{M} \setminus \mathcal{S}$, where $\mathcal{S}$ is a random subset of 30 Pauli measurements. Then, we average the fidelity over all possible test states.

The results are summarized in Table 1. Each row shows the performances of one particular trained GQNQ when tested using the measurement data from (i) 150 ground states of Ising model with $J \in \{0.1, ..., 1.5\}$, (ii) 150 ground states of Ising model with $J \in \{-1.5, -1.4, ..., -0.1\}$, where 10 test states are generated per value of $J$,

(iii) 10 ground states of Ising model with $J = 0$, (iv) 190 ground states of XXZ model with $\Delta \in \{-0.9, -0.8, ..., 0.9\}$, (v) 100 ground states of XXZ model with $\Delta \in \{-1.5, -1.4, ..., -1.1\} \cup \{1.1, 1.2, ..., 1.5\}$, where 10 test states are generated per value of $\Delta$, (vi) all the states from (i) to (v), (vii) 200 locally rotated GHZ states (viii) 200 locally rotated W states (vii), (ix) all the states from (i) to (v), together with (vii) and (viii). In the second column, the input data given to GQNQ is the true probability distribution computed with the Born rule, while in the third and fourth columns, the input data given to GQNQ during test is the finite statistics obtained by sampling the true outcome probability distribution 50 times and 10 times, respectively.

The results shown in Table 1 indicate that the performance with finite statistics is only slightly lower than the performance in the ideal case. It is also worth noting that GQNQ maintains a high fidelity even when used on multiple types of states.

Recall that the results in Table 1 refer to the scenario where GQNQ is trained with the full set of six-qubit Pauli measurements, which is informationally complete. An interesting question is whether the learning performance would still be good if the training used a non-informationally complete set of measurements. In Supplementary Note 4, we show that fairly accurate predictions can be made even if $\mathcal{M}$ consists only of 72 randomly chosen Pauli measurements.

While GQNQ makes accurate predictions for state families with sufficient structure, it should not be expected to work universally well on all possible quantum states. In Supplementary Note 3, we considered the case where the network is trained and tested on arbitrary six-qubit states, finding that the performance of GQNQ drops drastically. In Supplementary Note 5, we also provide numerical experiments on the scenario where some types of states are overrepresented in the training phase, potentially causing overfitting when GQNQ is used to characterize unknown states of an underrepresented type.

We now consider multiqubit states with 10, 20, and 50 qubits, choosing the measurement set $\mathcal{M}$ to consist of all two-qubit Pauli measurements on nearest-neighbor qubits and $\mathcal{S}$ a subset containing $s = 30$ measurements randomly chosen from $\mathcal{M}$. Here the dimension of state representation **r** is chosen to be 24, which guarantees a good performance in our numerical experiments.

For the Ising model, we choose the coupling between each nearest-neighbor pair of spins to be either consistently ferromagnetic for $J \geq 0$ or consistently antiferromagnetic for $J < 0$: for $J \geq 0$ we replace each coupling $J_i$ in Eq. (1) by $|J_i|$, and for $J < 0$ we replace $J_i$ by $-|J_i|$. The results are illustrated in Fig. 2. The figure shows that the average classical fidelities in both ferromagnetic and antiferromagnetic regions are close to one, with small drops around the phase transition point $J = 0$. The case where both ferromagnetic and anti-ferromagnetic interactions are present is studied in Supplementary Note 6, where we observe that the learning performance is less satisfactory in this scenario.

For XXZ model, the average classical fidelities in the XY phase are lower than those in the ferromagnetic interaction region, which is reasonable due to higher quantum fluctuations in the XY phase[35]. At the phase transition points $\Delta = \pm 1$, the average classical fidelities drop more significantly, partly because the abrupt changes of ground state properties at the critical points make the quantum state less predictable, and partly because the states at phase transition points are less represented in the training data set.

## Quantum state learning on a harmonic oscillator
We now test GQNQ on states encoded in harmonic oscillators, i.e. continuous-variable quantum states, including single-mode Gaussian states, as well as non-Gaussian states such as cat states and GKP states[40], both of which are important for fault-tolerant quantum computing[40,41]. For the measurement set $\mathcal{M}$, we choose 300 homodyne measurements, that is, 300 projective measurements associated to quadrature operators of the form $(e^{i\theta} \hat{a}^{\dagger} + e^{-i\theta} \hat{a})/2$, where $\hat{a}^{\dagger}$ and $\hat{a}$

a

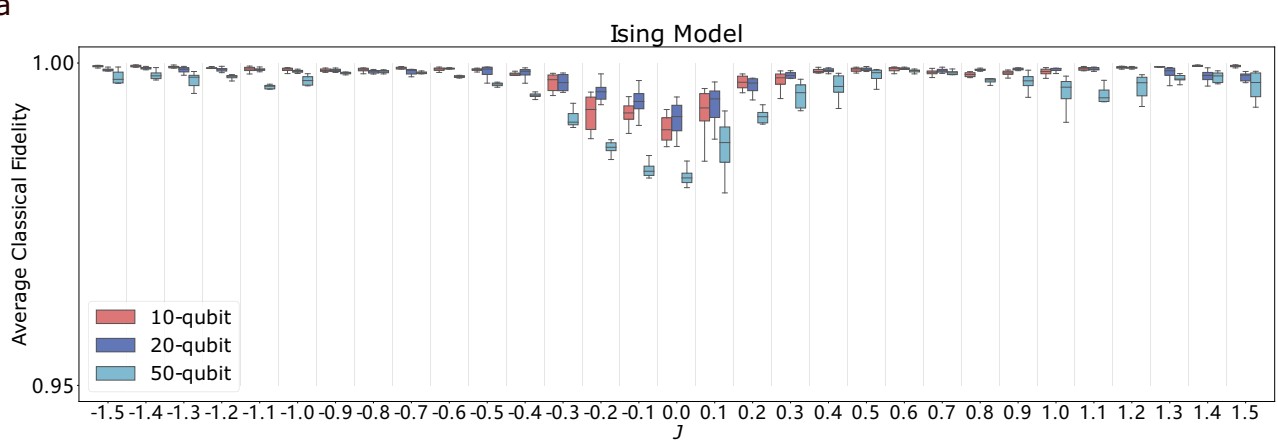

b

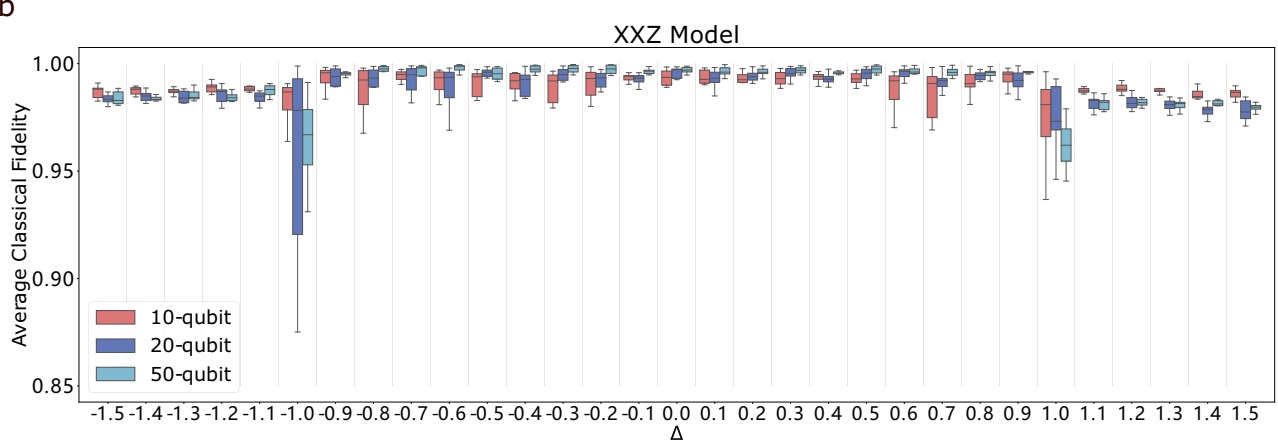

**Fig. 2 | Performance of GQNQs on Ising model ground states and XXZ model ground states visualized by boxplots[49].** Figure **a** shows the average classical fidelities of predictions given by three GQNQs for ten-, twenty- and fifty-qubit ground states of Ising model (1), respectively, with respect to different values of $J \in \{-1.5, -1.4, ..., 1.5\}$. Figure **b** shows the performance of another three GQNQs for ten-, twenty- or fifty-qubit ground states of XXZ model (2), respectively, with respect to different values of $\Delta \in \{-1.5, -1.4, ..., 1.5\}$. Given outcome probability distributions for all $\mathbf{m} \in \mathcal{S}$, each box shows the average classical fidelities of predicted outcome probabilities, averaged over all measurements in $\mathcal{M} \setminus \mathcal{S}$, for 10 instances.

**Table 2 | Performances of GQNQ on continuous-variable quantum states**

| Type of states for training and test | i. noiseless | worst case for i | ii. $\sigma(\text{noise}) = 0.05$ | worst case for ii | iii. $\sigma(\theta) = 0.05$ | worst case for iii |
|---|---|---|---|---|---|---|
| (i) Squeezed thermal states | 0.9973 | 0.9890 | 0.9964 | 0.9870 | 0.9972 | 0.9889 |
| (ii) Cat states | 0.9827 | 0.9512 | 0.9674 | 0.9053 | 0.9822 | 0.9461 |
| (iii) GKP states | 0.9762 | 0.9405 | 0.9746 | 0.9359 | 0.9758 | 0.9405 |
| (iv) (i)–(iii) together | 0.9658 | 0.9077 | 0.9264 | 0.8387 | 0.9643 | 0.9030 |

are bosonic creation and annihilation operators, respectively, and $\theta$ is a uniformly distributed phase in the interval $[0, \pi]$. For the subset $\mathcal{S}$, we pick 10 random quadratures. For the parametization of the measurements, we simply choose the corresponding phase $\theta$. Since the homodyne measurements have an unbounded and continuous set of outcomes, here we truncate the outcomes into a finite interval (specifically, at $\pm 6$) and discretize them, dividing the interval into 100 bins of equal width. The dimension of the representation vector $\mathbf{r}$ is chosen to be 16.

In Table 2 we illustrate the performance of GQNQ on (i) 200 squeezed thermal states with thermal variance $V \in [1, 2]$ and squeezing parameter $s$ satisfying $|s| \in [0, 0.5]$, $\arg(s) \in [0, \pi]$, (ii) 200 cat states corresponding to superpositions of coherent states with opposite amplitudes $|\alpha, \phi\rangle_{\text{cat}} := \frac{1}{\sqrt{\mathcal{N}}}(|\alpha\rangle + e^{i\phi}|-\alpha\rangle)$, where $\mathcal{N} = 2(1 + e^{-|\alpha|^2} \cos\phi)$, $|\alpha| \in [1, 3]$ and $\phi \in \{0, \frac{\pi}{8}, ..., \pi\}$, (iii) 200 GKP states that are superpositions of displaced squeezed states

$|\epsilon, \theta, \phi\rangle_{\text{gkp}} := e^{-\epsilon\bar{n}}(\cos\theta|0\rangle_{\text{gkp}} + e^{i\phi}\sin\theta|1\rangle_{\text{gkp}})$ where $\hat{n} = \hat{a}^\dagger\hat{a}$ is the photon number operator, $\epsilon \in [0.05, 0.2]$, $\theta \in [0, 2\pi]$, $\phi \in [0, \pi]$, and $|0\rangle_{\text{gkp}}$ and $|1\rangle_{\text{gkp}}$ are ideal GKP states, and (iv) all the states from (i), (ii), and (iii).

For each type of states, we provide the network with measurement data from $s = 10$ random homodyne measurements, considering both the case where the data is noiseless and the case where it is noisy. The noiseless case is shown in the second and third columns of Table 2, which show the classical fidelity in the average and worst-case scenario, respectively. In the noisy case, we consider both noise due to finite statistics, and noise due to an inexact specification of the measurements in the test set. The effects of finite statistics are modeled by adding Gaussian noise to each of the outcome probabilities of the measurements in the test. The inexact specification of the test measurements is modeled by rotating each quadrature by a random angle $\theta_i$, chosen independently for each measurement according to a

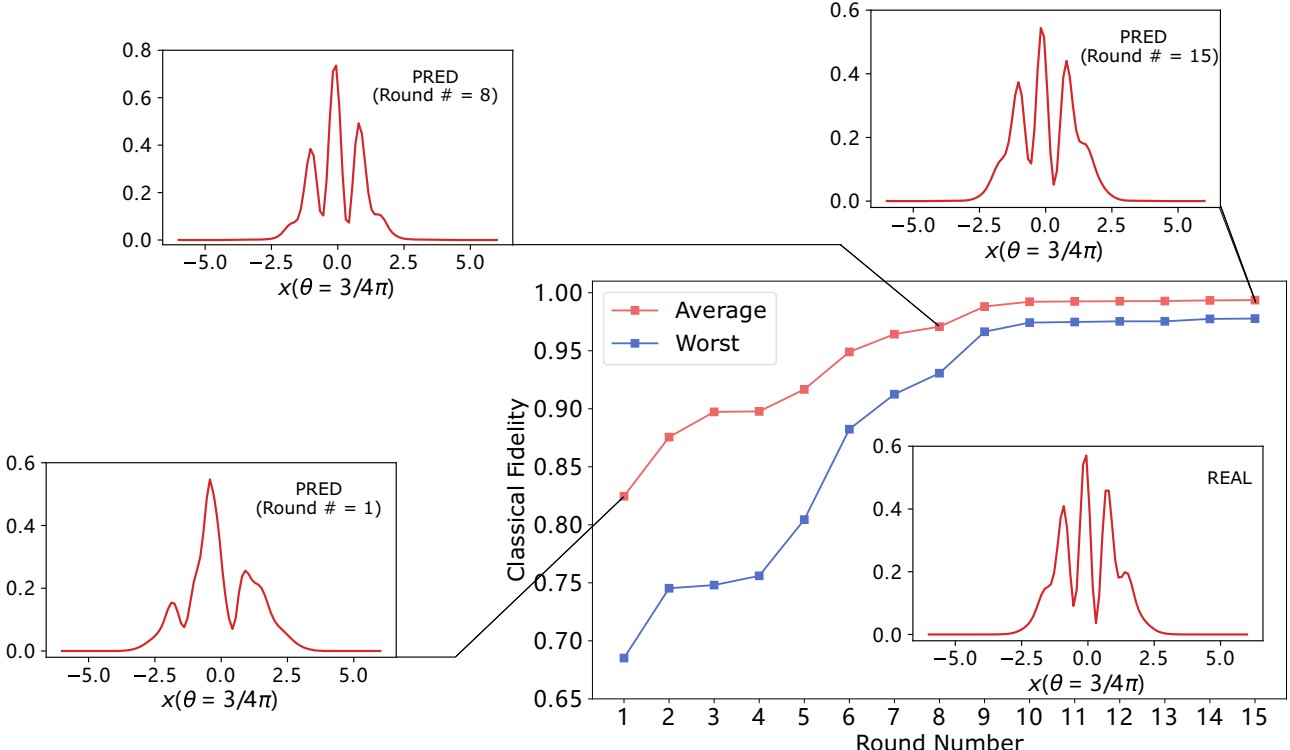

**Fig. 3 | Online learning of cat states in 15 time steps.** Each red point shows the classical fidelity, averaged over all the possible measurements in $\mathcal{M}$ and over all the cat states in the test set, which we take to be the same as in the experiments in the previous section. Each blue point is the worst-case classical fidelity over all possible query measurements, averaged over all the test states. Real outcome statistics and predicted outcome statistics at quadrature phase $\theta = 3/4\pi$ for an example cat state $|2.22 + 1.41i, \pi/4\rangle_{\mathrm{cat}}$ are plotted.

Gaussian distribution. The fourth and the fifth columns of Table 2 illustrate the effects of finite statistics, showing the classical fidelities in the presence of Gaussian added noise with variance 0.05. In the sixth and seventh columns, we include the effect of an inexact specification of the homodyne measurements, introducing Gaussian noise with variance 0.05. In all cases, the classical fidelity of predictions are computed with respect to the ideal noiseless probability distributions.

In Supplementary Note 6 we also provide a more detailed comparison between the predictions and the corresponding ground truths in terms of actual probability distributions, instead of their classical fidelities.

### Application to online learning

After GQNQ has been trained, it can be used for the task of online quantum state learning[13]. In this task, the various pieces of data are provided to the learner at different time steps. At the $t$-th time step, with $t \in \{1, ..., n\}$, the experimenter performs a measurement $\boldsymbol{M}_t$, obtaining the outcome statistics $\mathbf{p}_t$. The pair $(\mathbf{m}_t, \mathbf{p}_t)$ is then provided to the learner, who is asked to predict the measurement outcome probabilities for all measurements in the set $\mathcal{M} \setminus \mathcal{S}_t$ with $\mathcal{S}_t := \{\boldsymbol{M}_j\}_{j \leq t}$.

Online learning with GQNQ can be achieved with the following procedure. Initially, the state representation vector is set to $\mathbf{r}^{(0)} = (0, ..., 0)$. At the $t$-th time step, GQNQ computes the vector $\mathbf{r}_t = f_{\hat{\xi}}(\mathbf{m}_t, \mathbf{p}_t)$ and updates the state representation to $\mathbf{r}^{(t)} = [(t-1)\,\mathbf{r}^{(t-1)} + \mathbf{r}_t]/t$. The updated state representation is then fed into the generation network, which produces the required predictions. Note that updating the state representation does not require time-consuming operations, such as a maximum likelihood analysis. It is also worth noting that GQNQ does not need to store all the measurement data received in the past: it only needs to store the state representation $\mathbf{r}^{(t)}$ from one step to the next.

A numerical experiment on online learning of cat states is provided in Fig. 3. The figure shows the average classical fidelity at 15 subsequent time steps corresponding to 15 different homodyne measurements performed on copies of unknown cat states. The fidelity increases over time, confirming the intuitive expectation that the learning performance should improve when more measurement data are provided.

### Application to state clustering and classification

The state representation constructed by GQNQ can also be used to perform tasks other than predicting the outcome statistics of unmeasured POVMs. One such task is state clustering, where the goal is to group the representations of different quantum states into multiple disjoint sets in such a way that quantum states of the same type fall into the same set.

We now show that clusters naturally emerge from the state representations produced by GQNQ. To visualize the clusters, we feed the state representation vectors into a $t$-distributed stochastic neighbor embedding ($t$-SNE) algorithm[42], which produces a mapping of the representation vectors into a two-dimensional plane, according to their similarities. We performed numerical experiments using the types of six-qubit states in Table 1 and the types of continuous-variable states in Table 2. For simplicity, we restricted the analysis to state representation vectors constructed from noiseless input data.

The results of our experiments are shown in Fig. 4. The figure shows that states with significantly different physical properties correspond to distant points in the two-dimensional embedding, while states with similar properties naturally appear in clusters. For example, the ground states of the ferromagnetic XXZ model and the ground states in the gapless XY phase are clearly separated in Fig. 4a, in

a
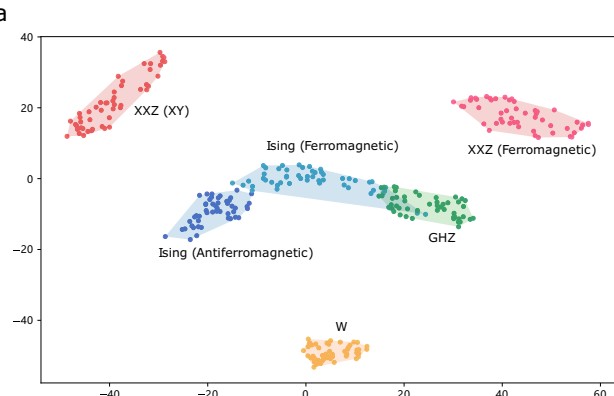
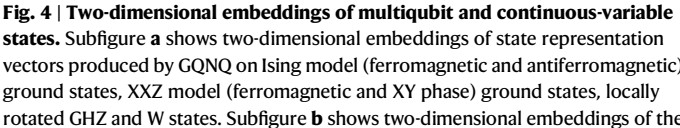

b
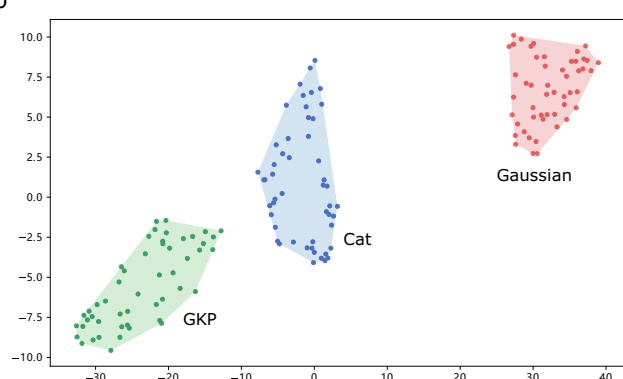

**Fig. 4 | Two-dimensional embeddings of multiqubit and continuous-variable states.** Subfigure **a** shows two-dimensional embeddings of state representation vectors produced by GQNQ on Ising model (ferromagnetic and antiferromagnetic) ground states, XXZ model (ferromagnetic and XY phase) ground states, locally rotated GHZ and W states. Subfigure **b** shows two-dimensional embeddings of the

state representation vectors of squeezed thermal states, cat states and GKP states. In both subfigures, shaded areas are added to help visualize the various type of states. Note that the representation vectors generated by GQNQ of states of the same type are near to each other in the two-dimensional embeddings.

agreement with the fact that there is an abrupt change of quantum properties at the phase transition point. On the other hand, in Fig. 4a, the ferromagnetic region of the Ising model is next to the anti-ferromagnetic region, both of which are gapped and short-range correlated. The ferromagnetic region of the Ising model appears to have some overlap with the region of GHZ states with local rotations, in agreement with the fact that the GHZ state is approximately a ground state of the ferromagnetic Ising model in the large $J$ limit.

The visible clusters in the two-dimensional embedding suggest that any unsupervised clustering algorithm could effectively cluster the states according to their representation vectors. To confirm this intuition, we applied a Gaussian mixture model[43] to the state representation vectors and chose the number of clusters to be equal to the actual number of state types (six for the six-qubit states, and three for the continuous-variable states). The portion of states whose types match the clusters is 94.67% for the six-qubit states, and 100% for the continuous-variable states.

The state representation produced by GQNQ can also be used to predict physical properties in a supervised model where an additional neural network is provided with labeled examples of states with a given property. In this setting, supervision can enable a more refined classification of quantum states, compared to the unsupervised clustering discussed before.

To illustrate the idea, we considered the problem of distinguishing between two different regimes in the Ising model, namely a regime where ferromagnetic interactions dominate ($J > 1$), and a regime both ferromagnetic and antiferromagnetic interactions are present ($0 < J < 1$). For convenience, we refer to these two regimes as to the pure and mixed ferromagnetic regimes, respectively. We use an additional neural network to learn whether a ground state corresponds to a Hamiltonian in the pure ferromagnetic regime or in the mixed one, using the state representation **r** of Ising ground states with ferromagnetic bias obtained from noiseless measurement data. The prediction reaches a success rate of 100%, 100%, and 99% for ten-qubit, twenty-qubit, and fifty-qubit ground states in our test sets, respectively. These high values can be contrasted with the clustering results in Fig. 4, where the pure ferromagnetic regime and the mixed one appear close to each other in the two-dimensional embedding.

## Discussion

Many works have explored the use of generative models for quantum state characterization[16,17,19–21], and an approach based on representation

learning was recently proposed by Iten et al[44]. The key difference between GQNQ and previous approaches concerns the training phase. In most previous works, the neural network is trained to reconstruct a single quantum state from experimental data. While this procedure can in principle be applied to learn any state, the training is state-specific, and the information learnt by the network through training on a given state cannot be automatically transferred to the reconstruction of a different quantum state, even if that state is of the same type. In contrast, the training of GQNQ works for multiple quantum states and for states of multiple types, thus enabling a variety of tasks, such as quantum state clustering and classification.

Another difference with previous works is that the training phase for GQNQ can use classically simulated data, rather than actual experimental data. In other words, the training can be carried out in an offline mode, before the quantum states that need to be characterized become available. By moving the training to offline mode, GQNQ can be significantly faster than other data-driven approaches that need to be trained with experimental data from unknown quantum states. The flip side of this advantage, however, is that offline training requires a partial supervision, which is not required in other state reconstruction approaches [16,17,19]. Indeed, the training of GQNQ requires quantum states in the same family as the tested state, and in order to implement the training offline one needs a good guess for the type of quantum state that will need to be characterized.

The situation is different if the training is done online, with actual experimental data provided from the quantum state to be characterized. In this setting, GQNQ behaves as a completely unsupervised learner that predicts the outcome statistics of unperformed measurements using measurement data obtained solely from the quantum state under consideration. Note that in this case the set of fiducial measurements $\mathcal{M}_*$ coincides with the set of performed measurements $\mathcal{S} \subset \mathcal{M}$. The details of the training procedure are provided in Supplementary Note 7. We performed numerical experiments in which GQNQ was trained with data from a single cat state, using data from 10 or 50 homodyne measurements. After the training, GQNQ was asked to predict the outcome statistics of a new randomly chosen homodyne measurement. The results are summarized in Table 3, where we show both the average classical fidelities averaged over all query measurements and worst-case classical fidelities over all query measurements.

Finally, we point out that our learning model shares some conceptual similarity with Aaronson's "pretty good tomography"[11], which

**Table 3 | Performances of GQNQ on cat states as an unsupervised learner**

| State | s = 50 (avg) | s = 50 (worst) | s = 10 (avg) | s = 10 (worst) |
|---|---|---|---|---|
| $\lvert 2,0\rangle_{\text{cat}}$ | 0.9918 | 0.9614 | 0.9912 | 0.9610 |
| $\lvert 2,\pi/4\rangle_{\text{cat}}$ | 0.9917 | 0.9602 | 0.9745 | 0.9236 |
| $\lvert 2.22+1.41i,\pi/4\rangle_{\text{cat}}$ | 0.9779 | 0.9171 | 0.9671 | 0.9133 |

aims at producing a hypothesis state that accurately predicts the outcome probabilities of measurements in a given set. While in pretty good tomography the hypothesis state is a density matrix, the form of the state representation in GQNQ is determined by the network itself. The flexibility in the choice of state representation allows GQNQ to find more compact descriptions for sufficiently regular sets of states. On the other hand, pretty good tomography is in principle guaranteed to work accurately for arbitrary quantum states, whereas the performance of GQNQ can be more or less accurate depending on the set of states, as indicated by our numerical experiments. An important direction of future research is to find criteria to determine a priori which quantum state families can be learnt effectively by GQNQ. This problem is expected to be challenging, as similar criteria are still lacking even in the original application of generative query networks to classical image processing.

## Methods

### Data generation procedures

Here we discuss the training/test dataset generation procedures. In the numerical experiments for ground states of Ising models and XXZ models, the training set is composed of 40 different states for each value of $J$ and $\Delta$, while the test set is composed of 10 different states for each value of $J$ and $\Delta$. For GHZ and W states with local rotations, we generate 800 states for training and 200 states for testing.

In the continuous-variable experiments, we randomly generate 10000 different states for each of the three families of squeezed thermal states, cat states, and GKP states. We then split the generated states into a training set and testing set, with a ratio of 4 : 1.

In the testing stage, the noiseless probability distributions for one-dimensional Ising models and XXZ models are generated by solving the ground state problem, either exactly (in the six qubit case) or approximately by density-matrix renormalization group (DMRG) algorithm[45] (for 10, 20, and 50 qubits). The data of continuous-variable quantum states are generated by simulation tools provided by Strawberry Fields[46].

### Network training

The training data set of GQNQ includes measurement data obtained from a fiducial set of quantum measurements $\mathcal{M}_*$ performed on a fiducial set of quantum states $\mathcal{Q}_* = \{(\sigma^{(k)})\}_{k=1}^{N}$. The fiducial states are divided into $N/B$ batches of $B$ states each. For each state in the $b$-th batch, a subset of fiducial measurements $\mathcal{M}_{\text{train}}^{(b)} \subset \mathcal{M}_*$ is randomly picked, and the network is provided with all the pairs $(\mathbf{m}, \mathbf{p}^{(k)})$, where $\mathbf{m}$ is the parametrization of a measurement in $\mathcal{M}_{\text{train}}^{(b)}$ and $\mathbf{p}^{(k)}$ is the corresponding vector of outcome probabilities on the state $\sigma^{(k)}$. The network is then asked to make predictions on the outcome probabilities of the rest of the measurements in $\mathcal{M}_* \setminus \mathcal{M}_{\text{train}}^{(b)}$, and the loss is computed from the difference between the real outcome probabilities (computed with the Born rule) and GQNQ's predictions (see Supplementary Note 1 for the specific expression of the loss function). For each batch, we optimize the parameters $\boldsymbol{\xi}$ and $\boldsymbol{\eta}$ of GQNQ by updating them along the opposite direction of the gradient of the loss function with respect to $\boldsymbol{\xi}$ and $\boldsymbol{\eta}$, using Adam optimizer[47] and batch gradient descent. The pseudocode for the training algorithm is also provided in Supplementary Note 1.

The training is repeated for $E$ epochs. In each epoch of the training phase, we iterate the above procedure over the $N/B$ batches of training data. For the numerical experiments in this paper, we typically choose $B = 30$ and $E = 200$.

### Network testing

After training, the parameters $\boldsymbol{\xi}$ and $\boldsymbol{\eta}$ are fixed. Then, the performance of GQNQ is tested on a set of test states $(\rho^{(j)})_{j=1}^{K}$. For each test state $\rho^{(j)}$, we randomly select an $s$-element subset $\mathcal{S}^{(j)}$ from the set $\mathcal{M}$ of possible POVM measurements. We then input the measurement data to the trained representation network, generate the state representation $\mathbf{r}^{(j)}$, and feed $\mathbf{r}^{(j)}$ into the trained generation network, asking it to predict the outcome probabilities for all the measurements in $\mathcal{M} \setminus \mathcal{S}^{(j)}$. Then we calculate the classical fidelity between each output prediction and the corresponding ground truth, and we average the fidelity over all possible measurements in $\mathcal{M} \setminus \mathcal{S}^{(j)}$.

### Hardware

Our neural networks are implemented by the pytorch[48] framework and trained on four NVIDIA GeForce GTX 1080 Ti GPUs.

## Data availability

The training and test data generated in this study have been deposited in the Figshare database, which can be accessed by https://doi.org/10.6084/m9.figshare.21211283.v2.

## Code availability

The codes that support the findings of this study are available in https://github.com/yzhuqici/GQNQ.

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

## Acknowledgements

This work was supported by funding from the Hong Kong Research Grant Council through grants no. 17300918 and no. 17307520, through the Senior Research Fellowship Scheme SRFS2021-7S02, the Croucher Foundation, and by the John Templeton Foundation through grant 61466, The Quantum Information Structure of Spacetime (qiss.fr). Y.X.W. acknowledges funding from the National Natural Science Foundation of China through grants no. 61872318. Research at the Perimeter Institute is supported by the Government of Canada through the Department of Innovation, Science and Economic Development Canada and by the Province of Ontario through the Ministry of Research, Innovation and Science. The opinions expressed in this publication are those of the authors and do not necessarily reflect the views of the John Templeton Foundation.

## Author contributions

Y.Z., Y.-D.W., and G.C. established the key idea in the paper. Y.Z. developed the neural network model and did the numerical experiments. Y.-D.W. wrote the draft paper. G.B., D.-S.W., and Y.W. contributed to the design and the implementation of numerical experiments. All the authors contributed to the preparation of the paper.

## Competing interests

The authors declare no competing interests.
