## [Peer Review File · Nature Communications]

Flexible learning of quantum states with generative query neural networksREVIEWER COMMENTS

Reviewer #1 (Remarks to the Author):

Quantum state reconstruction is a fundamental challenge in quantum technology. Recently several machine learning approaches have been applied to the problem of quantum state tomography (QST) as this is essentially a data-driven problem. In a seminal paper Torlai et. al. proposed a first implementation using neural-networks, more specifically restricted Boltzmann machines. They demonstrated the power of such unsupervised learning method reconstructing structured quantum states such as the W state or ground state of 2D Ising and Heisenberg hamiltonians, up to hundreds of sites.

In this manuscript, the authors propose a framework that should be more general, and in particular they claim that it is possible to overcome the necessity to re-train the model if the target state is changed.

I have some difficulties in understanding why this idea should even work in principle, as it appears to be counterintuitive.

For instance, the success of the neural-network approach stems from the high representative power of artificial neural networks to approximate structured quantum state, while maintaining the number of parameters reasonably low. As expected the approach of Torlai et. al. does not work for random states.

Here instead, I cannot find any physical argument that accompanies the presentation of the results. The authors only present relatively successful cases, while never providing examples where the method fails.

What would happen if one puts in the training dataset measurements pertaining to ground states coming from different instances of a frustrated Ising model (the couplings in Eq. 1 are only ferromagnetic)?

How would the training be possible if the networks are exposed to conflicting information?

Other very important discussions which are not present in the paper are:

(i) How do we know when the method fails? (ii) if a dataset is made of measurements coming from different quantum states I would expect overfitting to be a serious problem. What if the dataset contains an unbalanced number of type A states w.r.t B,C..etc..

The problem of overfitting becomes important when the true state is not known, which is of course the situation of interest.

Also, why the missing entries in Table I?

Minor comments:

I would recommend making also an example for the more usual case of discrete models.

I'm pretty sure that "the quadrature phase of a homodyne measurement" is not the simplest example to help a non-expert reader to understand the notation at a first glance.

The presentation of the method could be improved. It would be beneficial to have some toy examples to familiarise with the symbol's definition, such as the " m ".

Reviewer #2 (Remarks to the Author):

The current paper proposes using generative query neural networks to learn a representation of a quantum state. This representation is subsequently used to predict probabilities of unseen POVMs on the unknown state. In this sense, and also as noted by the authors, the learning task at hand is in the vein of Aaronson's 'pretty good tomography', rather than full state tomography, where the entire

density matrix of the state must be reconstructed.

I find the results very compelling and the diagrams/tables well-chosen. I recommend acceptance. I particularly enjoyed section D. That said, I have three minor comments and one major comment, which is more like a suggestion for an extension of this research.

My minor comments are the following: 1) in Table I, I think the first column should have the label 'training data'. Furthermore, the authors should also explain why they do not consider the case of noisy training data when multiple state types are included. 2) In several places throughout the manuscript, the authors write 'state' when they actually mean 'type of state' and this is rather confusing. For example in the conclusion, "In these works, a neural network is trained to reconstruct a particular experimental state, and hence, the trained network cannot be directly transferred to the reconstruction of a different experimental state." 3) As the author of reference [24], I disagree with your claim that our method only works for particular classes of states. In fact, we expressly say that there is no need to have any prior on the state class beforehand as the method reconstructs the full density matrix. Nevertheless, what I found unique about your method (that we do not do) is the aspect of learning only a prediction function for the acceptance probability of POVMs, which is potentially easier than learning the full state.

This brings me to my major comment. The main motivation for Aaronson proposing 'pretty-good tomography' was the intriguing observation that, for such a relaxation of full state tomography, the sample complexity (the number of copies of the state that one needs to prepare) only scales linearly in the number of qubits. This is of course an exponential improvement over full state tomography which requires 2^{2n} samples. However, as far as I am aware, these relationships only hold in theory-land. I am really curious about the following: do you actually observe this improvement empirically? If so, that would be an unprecedented bridge between quantum learning theory and practice. I have ideas for how to test this and would be interested to discuss this further.

Reviewer #3 (Remarks to the Author):

The authors propose a machine-learning method for the problem of quantum state learning based on a generative query network. The main advantage of this approach, the authors claim, in front of other existing ML-based methods in the literature is the ability to learn from measurement data coming from different quantum states in a given class, and successfully predict outcome probabilities of future measurements over multiple unknown states. This is an attractive feature which makes this work stand out among other approaches on quantum state learning. This feature is achieved by a combination of a representation network, which learns an averaged state representation r , and a generation network which uses this r to make predictions of a given measurement (not performed in the training step).

In my opinion, the authors' proposed ML model has great potential as a flexible platform for state learning that could be useful as part of an experimental workflow, especially by its ability to function in an online manner. However, I find that the manuscript is not sufficiently clear in several technical aspects, which prevents a fair evaluation of the protocol's actual usefulness and shortcomings. Therefore, I cannot make a recommendation to publish the manuscript in its current state, but I remain open to reconsider this if the authors can comment on the following points and update the manuscript accordingly.

On the representation networks and multiple states:

- 1) When trained over different states, the representation phase seems to produce an r which would correspond to the representation of the average state (by the authors' choice of the aggregate function A). Have you checked whether there is a difference between the setting in the manuscript and feeding training data coming directly from the average state?
- 2) If the above observation is correct, would using training data coming from orthogonal states (let's

fix $d=2$) produce a representation of the completely mixed state? This could still result in a good average fidelity (over states and measurements), but a very bad one for each chosen state. If this is true, this is a (maybe unavoidable?) shortcoming of the learning model that should be discussed.

3) In Section D, it is not clear to me what is the test input of the supervised model. Is it another average representation r' stemming from measuring a test quantum state? More details here would be welcomed.

On an adversarial setting:

4) The authors claim that the learning model works also in a setting where the query measurements are chosen by an adversary, instead of randomly. Yet, the numerical experiments seem to only show results for averaged queries of measurements (over the complementary set of the ones used for training). It would be interesting to see performance results on worst-case query measurements.

5) In addition, it would be useful to analyze how this worst-case or adversarial performance depends on the class of states used for training (e.g. single state, vs multiple states spanning a subspace of dimension $s < d$, vs multiple states spanning the whole Hilbert space).

On training:

6) In several places, it is said that only a representation m_i of the POVMs used is necessary for the model to work, instead of a full description of the operators in the set M_i , and that this can be considered an encrypted representation. Do you mean that you need less parameters? Otherwise, what would be the difference between a parametrization m_i (say the Bloch vectors associated to the POVM elements, which fully determine M_i) and a "full description" of M_i ?

7) In the supplementary material, the authors say that they "believe" that a recurrent neural network would yield better results instead of computing the average representation r . The authors should substantiate this claim, if only at an intuitive level.

8) Crucially, I could not find: the size of the training set used in the numerical experiments, the total number of parameters of the networks used, and the number of repetitions per measurement. These numbers comprise the minimal information needed by the reader to replicate the reported findings, and to quantitatively compare the resources required with other approaches.

On noise:

9) In Table 1, fidelities are reported for various cases considering noiseless and noisy data. However, noise is only considered for the scenarios with more than one state. Why is this? Are the results bad (in which case the numbers should still be reported), or is there a fundamental limitation preventing this analysis?

10) If there were noise not only in the measurement data but on the description (characterization) of the measurement devices, i.e., the m_i , how would this affect the performance? Is the model robust in this sense?

On generalization performance:

11) I would have liked to see a detailed discussion on generalization performance. Questions that come to mind: How does the richness of the measurement class considered affect it? If the measurement class is not informationally complete (which is mentioned in the introduction as a case where the model still works), does it generalize worse as one would expect? What is the effect in this regard of having an informationally incomplete class but using multiple states for training, vs using only one state?

We thank the Referees for their time and consideration. In the revised version, we addressed all the comments and suggestions raised by the Referees, carrying out further numerical experiments to test the performance of our neural network model in various physical scenarios.

The Referees' comments are reproduced verbatim and in full below as indented blue font, and our point-by-point responses are in non-indented black font. Wherever a change applied to the manuscript and we quote the exact change in the revised version, we use red font. At the end of the response, we list the main changes made in the new manuscript.

RESPONSE TO REVIEWER 1

Quantum state reconstruction is a fundamental challenge in quantum technology. Recently several machine learning approaches have been applied to the problem of quantum state tomography (QST) as this is essentially a data-driven problem. In a seminal paper Torlai *et. al.* proposed a first implementation using neural-networks, more specifically restricted Boltzmann machines. They demonstrated the power of such unsupervised learning method reconstructing structured quantum states such as the W state or ground state of 2D Ising and Heisenberg hamiltonians, up to hundreds of sites.

We agree with the Referee's summary about the importance of quantum state reconstruction and the essential role of data-driven approaches in this task, and we agree that Torlai *et al.*'s paper is a seminal work on the application of machine learning techniques to quantum state reconstruction.

In this manuscript, the authors propose a framework that should be more general, and in particular they claim that it is possible to overcome the necessity to re-train the model if the target state is changed.

We agree with this concise summary of the key contribution of our work.

I have some difficulties in understanding why this idea should even work in principle, as it appears to be counterintuitive. For instance, the success of the neural-network approach stems from the high representative power of artificial

neural networks to approximate structured quantum state, while maintaining the number of parameters reasonably low.

We appreciate that the results of our work may sound counterintuitive. In the revised version we have made an effort to provide more intuition as to why one might expect our approach to work, as well as more numerical experiments to support such intuition.

At the structural level, the key feature of our approach is to combine the power of deep representation learning and generative models. In GQNN, a representation network learns a mapping from measurement data to the representation of a quantum state, which is typically lower dimensional for sufficiently structured states. This mapping is expected to capture the main features of the state and map dissimilar states to dissimilar representation vectors. This data-driven state representation is then fed to a generation network that predicts the outcome probability distribution of a specified set of measurements. In contrast, the approach of restricted Boltzmann machine does not have a representation network: it only has a generation network that fits the measurement data from a given quantum state. This structural difference gives a first intuition of how our method can successfully learn multiple quantum states without being re-trained (provided that the quantum states to be learnt share some common structures with the states used in the training, such as being ground states of Hamiltonians in the same phase of quantum matter).

Further intuition comes from classical problem where generative query networks have been successfully used in the past years. GQNN is the quantum version of the generative query networks used in classical image processing to learn 3D scenes from 2D pictures. Formally, this task exhibits some analogies with the task of quantum state learning, where the outcome probabilities for a given quantum measurement can be viewed as a lower-dimensional projection of the density matrix. The success of generative query networks in reconstructing 3D scenes from their 2D projections suggests that GQNN should be similarly successful in reconstructing quantum states from their measurement statistics. The numerical experiments provided in the revised version of the paper provide a rather solid evidence that this intuition is indeed correct.

In the revised version, the above intuitions are included in the main text. In the section “The GQNN model”, we now write:

Our model of learner, GQNN, is a neural network composed of two main parts: a rep-

resentation network [1], producing a data-driven representation of quantum states, and a generation network [2], making predictions about the outcome probabilities of quantum measurements that have not been performed yet. The combination of a representation network and a generation network is called a generative query network [3]. This type of neural network was originally developed for the classical task of learning 3D scenes from 2D snapshots taken from different viewpoints. The intuition for adapting this model to the quantum domain is that the statistics of a fixed quantum measurement can be regarded as a lower-dimensional projection of a higher-dimensional object (the quantum state), in a way that is analogous to a 2D projection of a 3D scene. The numerical experiments reported in this paper indicate that this intuition is indeed correct, and that GQNQ works well even in the presence of errors in the measurement data and fluctuations due to finite statistics.

[1] Bengio, Courville & Vincent, "Representation learning: A review and new perspectives", *IEEE Trans. Pattern Anal. Mach. Intell.* 35, 1798 (2013)

[2] David Foster, *Generative deep learning: teaching machines to paint, write, compose, and play* (O'Reilly Media, 2019)

[3] M Ali Eslami, Danilo Jimenez Rezende, Frederic Besse, Fabio Viola, Ari S Morcos, Marta Garnelo, Avraham Ruderman, Andrei A Rusu, Ivo Danihelka, Karol Gregor, et al., "Neural scene representation and rendering," *Science* 360, 1204–1210 (2018).

In the revised version we also mention that our representation network is not constrained to a given choice of state representation, such as the density matrix or the wavefunction used in restricted Boltzmann machines. Instead, GQNQ has the flexibility to build its own data-driven representations of quantum states, which can capture in a more concise way the structure of certain subsets of quantum states. In the subsection "The GQNQ network" of Results, we added the following comments:

While tomographic protocols strive to find the density matrix that fits the measurement data, GQNQ is not constrained to a specific choice of representation. This additional freedom enables the network to construct a lower-dimensional representation of quantum states with sufficiently regular structure, such as ground states in well-defined phases of matter, and to make predictions for states that did not appear in the training phase.

As expected the approach of Torlai et. al. does not work for random states. Here instead, I cannot find any physical argument that accompanies the presentation

of the results.

Thank you for raising this point; we agree that it should be discussed in the paper. It is true that the approach of Torlai et al. does not work well for arbitrary quantum states. More generally, this issue affects most deep learning approaches to quantum state reconstruction. In this respect, GQNN is no exception: it is effective in characterizing well-structured quantum states, but gives less satisfactory results for states with less structure. In the revised version, we explicitly mention that our approach should not be expected to work well for arbitrary quantum states, and we provide explicit examples of cases where the performance drops for certain hard-to-learn states, giving physical arguments for why GQNN performs better on some states and less well on others.

In the subsection “Quantum state learning in spin systems” of Results, we added the following sentences to clarify that the model cannot be used to learn a random quantum state.

While GQNN makes accurate predictions for state families with sufficient structure, it should not be expected to work universally well on all possible quantum states. In Supplementary Note 3, we considered the case where the network is trained and tested on arbitrary six-qubit states, finding that the performance of GQNN drops drastically.

In the same subsection, for ground states of the XXZ spin model, we added the following discussion about the performances of our neural network model in different quantum phases and especially at phase transition points.

For XXZ model, the average classical fidelities in the XY phase are lower than those in the ferromagnetic interaction region, which is reasonable due to higher quantum fluctuations in the XY phase. At the phase transition points $\Delta = \pm 1$, the average classical fidelities drop more significantly, partly because the abrupt changes of ground state properties at the critical points make the quantum state less predictable, and partly because the states at phase transition points are less represented in the training data set.

Physical arguments explaining why our approach works well for certain sets of states and less well for other sets of states are provided also in relation to the task of quantum clustering: in the subsection “Application to state clustering and classification” of Results, we added the following discussion:

The figure shows that states with significantly different physical properties correspond

to distant points in the two-dimensional embedding, while states with similar properties naturally appear in clusters. For example, the ground states of the ferromagnetic XXZ model and the ground states in the gapless XY phase are clearly separated in Fig. 4(a), in agreement with the fact that there is an abrupt change of quantum properties at the phase transition point. On the other hand, in Fig. 4(a), the ferromagnetic region of the Ising model is next to the antiferromagnetic region, both of which are gapped and short-range correlated. The ferromagnetic region of the Ising model appears to have some overlap with the region of GHZ states with local rotations, in agreement with the fact that the GHZ state is approximately a ground state of the ferromagnetic Ising model in the large J limit.

The authors only present relatively successful cases, while never providing examples where the method fails.

We agree that the examples presented in the original version of the manuscript did not provide enough information about the cases where our method does not perform satisfactorily. In the revised version, several “unsuccessful examples” are included. In particular, we study the performance of GQNN for random six-qubit states and show that the performance of GQNN drops drastically, indicating GQNN cannot be used to learn arbitrary quantum states. Another example where our approach does not perform well is at phase transition points of ground states of the XXZ model, as shown by our experiments for 10, 20, and 50 qubits.

What would happen if one puts in the training dataset measurements pertaining to ground states coming from different instances of a frustrated Ising model (the couplings in Eq. 1 are only ferromagnetic)?

Thank you for the interesting question. In the revised version, we added a discussion of this scenario. For 6 qubits, with all Pauli measurements in the training set, we find that GQNN maintains a good performance, with classical fidelities close to 1, even in the presence of both ferromagnetic and anti-ferromagnetic couplings. For 10-, 20- and 50-qubit ground states, instead, we considered the case where only nearest neighbor Pauli measurements are used. In this scenario, we found out that the presence of both ferromagnetic and antiferromagnetic interactions results in ground states that are not sufficiently well-structured to be

learnt only from the data of nearest neighbor measurements. The details of the analysis are presented in Supplementary Note 6.

Besides the Ising model, we also added new results on the one-dimensional XXZ model. In this case, we find that the performance of GQNN in the XY phase is a bit lower than in the ferromagnetic phase, which is reasonable due to high quantum fluctuations in XY phase that make the states harder to learn.

How would the training be possible if the networks are exposed to conflicting information?

The answer to this question depends on which kind of conflicting information we are considering. If we refer to training data from states of different types, such as ferromagnetic and antiferromagnetic ground states, it appears that the representation network in GQNN is able to autonomously sort the training states, representing states of different types with significantly different vectors (this feature is exploited in our paper for the application to quantum state clustering). While this feature is quite noteworthy, it is worth stressing that in each step of training, GQNN receives measurement data from a given quantum state: even if the training states and their types are unknown, GQNN knows which data come from the same quantum state and which data come from different states.

On the other hand, if “conflicting information” means erroneous specifications of the measurements in the training set, or noisy data that may not correspond exactly to any quantum state, then the performance of GQNN will depend on the extent of the errors. The effects of different types of errors are analyzed in Tables I and II.

Other very important discussions which are not present in the paper are: (i)
How do we know when the method fails?

The safest scenario is probably the scenario where the quantum state to be characterized is known to belong to a family of well-structured states on which GQNN has been observed to work successfully, such as the families of states considered in the paper. More generally, since GQNN makes predictions about the statistics of measurements that have not been performed yet, a simple approach to check the reliability of these predictions is to perform a randomized test, comparing the predictions of GQNN with the actual statistics of random measurements performed on the state under consideration. Of course, it would be ideal to

have *a priori* criteria to determine in advance on which quantum states GQNQ will work well. However, such criteria are missing even in the original application of generative query networks to classical image processing, where the best results are obtained for 3D scenes that are sufficiently “well-structured,” but no mathematical guarantee of success is provided *a priori*.

(ii) if a dataset is made of measurements coming from different quantum states I would expect overfitting to be a serious problem. What if the dataset contains an unbalanced number of type A states w.r.t B,C..etc.. The problem of overfitting becomes important when the true state is not known, which is of course the situation of interest.

In the revised version, we included numerical experiments that test how the model works when the training data of multiple types of states are unbalanced. We considered the scenario in which the state to be characterized is of a type that is underrepresented in the training data (specifically, we set the type of the tested state to appear 10 times less frequently than the other types of states). The results show that the performance of GQNQ with unbalanced training data depends on the state under consideration. For W states with local rotations we find that unbalanced training data has little effect on the performance. The situation is similar for the ground states of the Ising model in the ferromagnetic phase. In contrast, the prediction for XXZ model in the XY phase drops to 0.73 when the training data are unbalanced. We have added one sentence in the subsection “Quantum state learning in spin systems” mentioning that we have studied this issue referring the reader to Supplementary Note 5 for the details.

Also, why the missing entries in Table I?

Thanks for spotting this issue. We have added the missing entries in the revised version.

Minor comments:

I would recommend making also an example for the more usual case of discrete models. I’m pretty sure that “the quadrature phase of a homodyne measurement” is not the simplest example to help a non-expert reader to understand the notation at a first glance.

Thank you for this suggestion, very much appreciated. In the revised version we have added to our examples various multiqubit states, including GHZ states with local rotations and W states with local rotations, ground states of the XXZ model including both ferromagnetic interaction and XY phase, and ground states of the antiferromagnetic Ising model (in addition to the ferromagnetic Ising ground states already present in the original version). The results on all these examples are presented in the subsection “Quantum state learning in spin systems”.

The presentation of the method could be improved.

We took this comment seriously, and thoroughly revised the presentation of our method in the Results section, and in the Methods section. In the current version, the main text explains the data generation procedures, network training and network testing in a more accessible way, suitable for non-expert readers, while the pseudocode and the more technical details are provided in Supplementary Note 1.

It would be beneficial to have some toy examples to familiarise with the symbol’s definition, such as the “m”.

Thanks for this wise suggestion. In the revised version, we added the following sentence in the subsection “Quantum state learning framework”.

Here the parametrization \mathbf{m}_i could be the full description of the POVM \mathbf{M}_i , or a lower-dimensional parametrization valid only for measurements in the set \mathcal{M} . For example, if \mathcal{M} contains measurements of linear polarization, a measurement in \mathcal{M} could be parametrized by the angle θ of the corresponding polarizer.

We also provided examples of the choice of parameters for the characterization of many-qubit states and continuous-variable states.

Many-qubit states: To parameterize the measurements in \mathcal{M} , we provide the entries in the corresponding Pauli matrix at each qubit, arranging the entries in a 48-dimensional real vector.

Continuous-variable states: For the measurement set \mathcal{M} , we choose 300 homodyne measurements, that is, 300 projective measurements associated to quadrature operators of the form $(e^{i\theta} \hat{a}^\dagger + e^{-i\theta} \hat{a})/2$, where \hat{a}^\dagger and \hat{a} are bosonic creation and annihilation operators, respectively, and θ is a uniformly distributed phase in the interval $[0, \pi)$. For the parametrization of the measurements, we simply choose the corresponding phase θ .

RESPONSE TO REVIEWER 2

The current paper proposes using generative query neural networks to learn a representation of a quantum state. This representation is subsequently used to predict probabilities of unseen POVMs on the unknown state. In this sense, and also as noted by the authors, the learning task at hand is in the vein of Aaronson’s ‘pretty good tomography’, rather than full state tomography, where the entire density matrix of the state must be reconstructed.

We agree with this insightful summary of the settings of our work.

I find the results very compelling and the diagrams/tables well-chosen. I recommend acceptance. I particularly enjoyed section D.

We thank the Referee for the positive assessment of our results and for the recommendation towards publication. We are pleased that the Referee found section D enjoyable.

That said, I have three minor comments and one major comment, which is more like a suggestion for an extension of this research.

We appreciate the Referee’s comments and suggestions, which helped us to further improve the manuscript.

My minor comments are the following: 1) in Table I, I think the first column should have the label ‘training data’. Furthermore, the authors should also explain why they do not consider the case of noisy training data when multiple state types are included.

Very good point, thanks for bringing it up. In the revised version, we have added “Types of states for training and test” in the first column of the table and also included all the noisy cases for quantum state learning of multiple state types.

2) In several places throughout the manuscript, the authors write ‘state’ when they actually mean ‘type of state’ and this is rather confusing. For example in the conclusion, “In these works, a neural network is trained to reconstruct a particular experimental state, and hence, the trained network cannot be directly transferred to the reconstruction of a different experimental state.”

Thank you for pointing out the issue. In the revised version, we carefully checked the use of the expressions “state” and “type of state” throughout the paper. In particular, the above sentence has been revised as follows:

While this procedure can in principle be applied to learn any state, the training is state-specific, and the information learnt by the network through training on a given state cannot be automatically transferred to the reconstruction of a different quantum state, even if that state is of the same type.

3) As the author of reference [24], I disagree with your claim that our method only works for particular classes of states. In fact, we expressly say that there is no need to have any prior on the state class beforehand as the method reconstructs the full density matrix. Nevertheless, what I found unique about your method (that we do not do) is the aspect of learning only a prediction function for the acceptance probability of POVMs, which is potentially easier than learning the full state.

We are sorry that our comment in the previous version of the manuscript could give a wrong impression. We definitely did not mean that Ref. [24] and other previous approaches were restricted to a particular class of states. What we meant is that there is a structural difference in the way the training is used. For GQNQ, after the training phase has been concluded and the parameters of the network have been fixed, the trained network can be used to predict outcome probability distributions for multiple quantum states that were not included in the training set, and can even be used to compare states of different types. This feature is different from those of most previous works, in which the networks are trained with data from the state to be characterized.

Regarding your second comment, we agree that the learning task in our manuscript is to predict the statistics of some selected POVMs, which can be simpler than learning the full description of a state when the set of selected POVMs is not informationally complete.

This brings me to my major comment. The main motivation for Aaronson proposing ‘pretty-good tomography’ was the intriguing observation that, for such a relaxation of full state tomography, the sample complexity (the number of copies of the state that one needs to prepare) only scales linearly in the number of

qubits. This is of course an exponential improvement over full state tomography which requires 2^{2n} samples. However, as far as I am aware, these relationships only hold in theory-land. I am really curious about the following: do you actually observe this improvement empirically? If so, that would be an unprecedented bridge between quantum learning theory and practice. I have ideas for how to test this and would be interested to discuss this further.

This is indeed a very promising research direction. To some extent, the numerical results in our paper do indeed suggest that for structured quantum states, like Hamiltonian ground states, predicting the statistics of a restricted set of measurements \mathcal{M} (e.g. nearest neighbor Pauli measurements) requires much smaller amount of data than the exponential data required for full tomography. Since the set of performed measurements \mathcal{S} is a subset of the set \mathcal{M} , every family of states where our method works well with a linear-size \mathcal{M} is trivially an example where one can achieve accurate predictions with a number of performed measurements that grows linearly or even sublinearly. On the other hand, it would be interesting to explore whether the set of performed measurements \mathcal{S} can be exponentially smaller than the set of possible measurements \mathcal{M} . The challenging part here is to provide evidence of good performance when the set \mathcal{M} is exponentially large, as computing the classical fidelity between the predictions of GQNN and all the correct statistics for all measurements in \mathcal{M} becomes unfeasible. In general, as you correctly point out, bridging the gap between our practical algorithm and Aaronson's general theory requires further research, which we would be very keen to discuss. Part of the challenge is that the performance of GQNN depends on the specific states under consideration. In a numerical experiment with 6 qubits, the results indicate that GQNN does not work universally for all quantum states (cf. Supplementary Note 3 in the revised version), in contrast with Aaronson's pretty good tomography which, at least in theory, is promised to work for arbitrary quantum states. Nevertheless, we do expect is that GQNN should achieve the benefits of pretty good tomography for well structured families of quantum states. In the revised version, this point is briefly touched upon in the Discussion section.

RESPONSE TO REVIEWER 3

The authors propose a machine-learning method for the problem of quantum state learning based on a generative query network. The main advantage of this approach, the authors claim, in front of other existing ML-based methods in the literature is the ability to learn from measurement data coming from different quantum states in a given class, and successfully predict outcome probabilities of future measurements over multiple unknown states. This is an attractive feature which makes this work stand out among other approaches on quantum state learning.

We thank the Referee for the accurate summary of the main feature of our work, and for the positive comments on its relevance to the state of the art in the field.

This feature is achieved by a combination of a representation network, which learns an averaged state representation r , and a generation network which uses this r to make predictions of a given measurement (not performed in the training step).

Thank you again for the accurate summary. In the revised version, we introduced a slight change of terminology, as we spotted a potential ambiguity in the notion of “averaged representation.” There, it is important to clarify that our state representation is *not* averaged over different quantum states, but only over different measurements performed on the same quantum state. To avoid a potential confusion, in the revised version we have replaced the name “averaged state representation” by “state representation,” and we stressed that the state representation is obtained by averaging vectors obtained from different measurement data on the same quantum state.

In my opinion, the authors’ proposed ML model has great potential as a flexible platform for state learning that could be useful as part of an experimental workflow, especially by its ability to function in an online manner.

We thank the Referee for this positive comment on our method and on its potential applicability to actual experimental workflows. This is indeed one of the aspects of our work about which we are most excited.

However, I find that the manuscript is not sufficiently clear in several technical aspects, which prevents a fair evaluation of the protocol’s actual usefulness and shortcomings. Therefore, I cannot make a recommendation to publish the manuscript in its current state, but I remain open to reconsider this if the authors can comment on the following points and update the manuscript accordingly.

We appreciate the Referee’s careful and balanced assessment, and we have made a major effort in improving the clarity of presentation. In the revised version, we have addressed all the Referee’s comments and suggestions, as summarized in the following point-by-point response.

On the representation networks and multiple states:

1) When trained over different states, the representation phase seems to produce an \mathbf{r} which would correspond to the representation of the average state (by the authors’ choice of the aggregate function A). Have you checked whether there is a difference between the setting in the manuscript and feeding training data coming directly from the average state?

This comment may have been resulted from the aforementioned ambiguity of the term “averaged representation.” In the revised version, we clarify that each vector \mathbf{r}_i ($1 \leq i \leq s$) refers to the same quantum state ρ , while the subscript i labels different measurements performed on this specific quantum state. One can think of \mathbf{r}_i as a candidate representation of the quantum state ρ , based on data from the i -th measurement. The aggregate function calculates the average of \mathbf{r}_i over all values of i , that is, it produces a state representation \mathbf{r} that takes into account the information about all the measurements that have been performed on the quantum state under consideration. The average over i yields a state representation \mathbf{r} of *this particular state* ρ . To train or test our model on multiple quantum states, we repeat this procedure, each time using the data from a different quantum state, but without taking an average over the various states under consideration.

2) If the above observation is correct, would using training data coming from orthogonal states (let’s fix $d=2$) produce a representation of the completely mixed state? This could still result in a good average fidelity (over states and measurements), but a very bad one for each chosen state. If this is true, this is a (maybe unavoidable?) shortcoming of the learning model that should be discussed.

We agree that, if this were the case, it would be a serious shortcoming. Fortunately, as it should be clear from the above explanation, our method does not suffer from this problem. Still, your comment made us fully appreciate the ambiguity of the term “averaged representation,” and in the revised version we paid extra care to spell out our procedure in a way that avoids the ambiguity and avoids the above concern.

3) In Section D, it is not clear to me what is the test input of the supervised model. Is it another average representation r' stemming from measuring a test quantum state? More details here would be welcomed.

Thank you for the suggestion. In the revised version, we added the following sentence to explain the state representations we use in subsection “Application to state clustering and classification” for the supervised model.

We use an additional neural network to learn whether a ground state corresponds to a Hamiltonian in the pure ferromagnetic regime or in the mixed one, using the state representation r of Ising ground states with ferromagnetic bias obtained from noiseless measurement data.

On an adversarial setting:

4) The authors claim that the learning model works also in a setting where the query measurements are chosen by an adversary, instead of randomly. Yet, the numerical experiments seem to only show results for averaged queries of measurements (over the complementary set of the ones used for training). It would be interesting to see performance results on worst-case query measurements.

This is an excellent point, thank you for bringing it up. In the revised version, we have added the worst-case classical fidelity over all possible measurements, showing that its value is comparable with that of the average classical fidelity in most of the examples studied in the paper.

5) In addition, it would be useful to analyze how this worst-case or adversarial performance depends on the class of states used for training (e.g. single state, vs multiple states spanning a subspace of dimension $s < d$, vs multiple states spanning the whole Hilbert space).

In the revised version, we observe the value of the worst-case performance of GQNG over examples of continuous-variable states in two different scenarios. Table II shows the worst-case performance over sets of test states when the network is trained with measurement data from a fiducial set of quantum states, and in contrast, Table III shows the worst-case performance when the network is trained with the measurement data of a single state to be characterized.

The results in Table II indicate that enlarging the classes of states for training and test results in a decrease of the worst-case fidelity, as it is intuitively reasonable because learning the features of different types of states simultaneously is harder than learning the feature of one type of states.

The results in Table III indicate that the worst-case performance for a single state depends on the state under consideration. For example, the worst-case classical fidelity for a cat state $|2, \pi/4\rangle_{\text{cat}}$ with complex superposition coefficients is lower than the worst-case classical fidelity for a special cat state $|2, 0\rangle_{\text{cat}}$ with real superposition coefficients.

On training: 6) In several places, it is said that only a representation m_i of the POVMs used is necessary for the model to work, instead of a full description of the operators in the set M_i , and that this can be considered an encrypted representation. Do you mean that you need less parameters? Otherwise, what would be the difference between a parametrization m_i (say the Bloch vectors associated to the POVM elements, which fully determine M_i) and a “full description” of M_i ?

Thank you for the insightful questions, which helped us clarify the point we wanted to make. The parametrization can be either a full description of each POVM operator or a lower-dimensional parameterization valid only for a restricted set of measurements, denoted by \mathcal{M} in the paper. For example, for a quadrature measurement, a simple parametrization is to specify the parameter $\theta \in [0, 2\pi)$ of the quadrature operator $(a^\dagger e^{i\theta} + a e^{-i\theta})/2$.

No matter whether the number of parameters is smaller or not, an important feature is that the parametrization could be encrypted, in the sense that it may not reveal what measurement is actually performed in the laboratory. For example, if we only give the angle θ to the network, the network will not know that this angle refers to a quadrature of a continuous variable system—the angle could also refer e.g. to the direction of a polarizer

for a measurement of polarization on a single photon. Moreover, one can also add further encryption operations, such as shifting the angle θ by an amount θ_0 . More generally, one could apply any continuous mapping from the original parameters in the usual description of measurement.

In the revised version, we added the following sentences in the subsection “Quantum state learning framework” to clarify this point.

Here the parametrization \mathbf{m}_i could be the full description of the POVM \mathbf{M}_i , or a lower-dimensional parametrization valid only for measurements in the set \mathcal{M} . For example, if \mathcal{M} contains measurements of linear polarization, a measurement in \mathcal{M} could be parametrized by the angle θ of the corresponding polarizer. The parametrization could also be encrypted, so that the actual description of the quantum hardware in the experimenter’s laboratory is concealed from the learner.

7) In the supplementary material, the authors say that they “believe” that a recurrent neural network would yield better results instead of computing the average representation \mathbf{r} . The authors should substantiate this claim, if only at an intuitive level.

There are two main reasons why we make such a statement. First, we can always regard a neural network as a mapping function. Hence, a fixed aggregate function, achieved by a particular recurrent neural network, provides a lower bound guarantee on the performance of that neural network. Since the average function used in our work is just one particular example of aggregate function, we believe that there exist other neural networks having the potential to achieve better performance. Second, we have observed that in the research of representation learning for classical data, such as representation learning of sentences [1] or graphs [2], trainable weighted aggregate functions can often yield better results. Therefore, we believe that the same effect may also take place in the representation learning of structured quantum states.

However, using a trainable neural network instead of a fixed aggregate function will inevitably increase the number of parameters in the model, which leads to higher requirements for hardware and hyperparameter tuning in the training, and increases the training time. For these reasons, here we just adopt the simplest choice of the average as the aggregate function in our model. In a future work, we are planning to explore more elaborate architectures in

more sophisticated learning scenarios, such as learning dynamic processes.

[1] Jacob Devlin, Ming-Wei Chang, Kenton Lee, Kristina Toutanova: BERT: Pre-training of Deep Bidirectional Transformers for Language Understanding. NAACL-HLT (1) 2019: 4171-4186

[2] Veličković, P., Cucurull, G., Casanova, A., Romero, A., Liò, P. and Bengio, Y. (2017). Graph Attention Networks. 6th International Conference on Learning Representations, .

8) Crucially, I could not find: the size of the training set used in the numerical experiments, the total number of parameters of the networks used, and the number of repetitions per measurement. These numbers comprise the minimal information needed by the reader to replicate the reported findings, and to quantitatively compare the resources required with other approaches.

We agree. In the revised version, we included the specification of all these numbers. The discussion about training data size is now included in the Methods section. We also include the number of repetitions per measurement in Table I and present more details about the numerical experiments in the Supplementary Note 1, including the number of trainable parameters, initialization of parameters, learning rate, number of epochs as well as training time.

On noise:

9) In Table 1, fidelities are reported for various cases considering noiseless and noisy data. However, noise is only considered for the scenarios with more than one state. Why is this? Are the results bad (in which case the numbers should still be reported), or is there a fundamental limitation preventing this analysis?

There was actually no particular reason for having the noisy results only in the multiple state cases, other than we had chosen that scenario as an example to illustrate the noise robustness. In the revised version, we included the noisy cases for all the examples of quantum state learning applications. In the continuous variable case, the effects of finite statistics are modelled by adding Gaussian noise to each of the outcome probabilities of the measurements in the test. We also study the noisy cases when GQNNQ is used to learn different types of six-qubit states in Table I, where the input data given to GQNNQ during test can be the finite statistics obtained by sampling the true outcome probability distribution

50 times or 10 times. The average classical fidelities in both Table I and Table II show that our model is robust to the stochastic noises of finite measurement statistics even when learning multiple types of states.

10) If there were noise not only in the measurement data but on the description (characterization) of the measurement devices, i.e., the m_i , how would this affect the performance? Is the model robust in this sense?

Thank you for the excellent question. In the revised version, we have added numerical experiments where there exists noise due to an inexact specification of the measurements in the test set, in the subsection “Quantum state learning on a harmonic oscillator”. The inexact specification of the test measurements is modelled by rotating each quadrature by a random angle θ_i , chosen independently for each measurement according to a Gaussian distribution. By including the effect of an inexact specification of the homodyne measurements, introducing Gaussian noise with variance 0.05, the results in Table II show that our model is robust to small errors in the specification of the measurements.

On generalization performance:

11) I would have liked to see a detailed discussion on generalization performance. Questions that come to mind: How does the richness of the measurement class considered affect it? If the measurement class is not informationally complete (which is mentioned in the introduction as a case where the model still works), does it generalize worse as one would expect? What is the effect in this regard of having an informationally incomplete class but using multiple states for training, vs using only one state?

Thank you also for this suggestion. In the revised version, we have done more numerical experiments in the examples of six-qubit quantum states and continuous-variable states to show the generalization performance of our model when measurement class is a much smaller set than an informationally complete set of measurements. In the new experiments on six-qubit states, we randomly select only 72 different six-qubit Pauli-basis measurements (vs 729 Pauli-basis measurements in the informationally complete scenario) as a fiducial set of measurements for training. In the new experiments on continuous-variable states, we use measurement data over only 10 homodyne measurements (vs 300 homodyne measurements

in the truncated space in informationally complete scenario) for training. In both cases, we find that the trained network still has great power in predicting outcome statistics for measurements not yet performed, with the average classical fidelities only a bit lower than those in the informationally complete scenario. This can be explained by the fact that these quantum states under consideration lie in lower dimensional corners of the entire Hilbert space, and can be effectively learned by our model from the measurement statistics of a set of measurements far from information completeness. We have included the discussions in the subsection “Quantum state learning in spin systems”. More details about the numerical results are presented in the Supplementary Note 4.

In addition, we also included a study of the single-state scenario where GQNN is trained with measurement data from a single state, namely the state that has to be characterized. We performed numerical experiments in which GQNN was trained with data from a single cat state, using data from 10 (informationally incomplete) or 50 (informationally complete) homodyne measurements. The results show that even in the informationally incomplete case, GQNN still has great power in prediction of outcome statistics of unperformed measurements. The detailed performance of GQNN for single cat state is shown in Table III.

List of Key Changes

1. In the new manuscript, we add discussions to clarify the point that our network is trained with data of a set of fiducial quantum states and after training, is used to characterize different quantum states that share common structure with the fiducial states.

2. For quantum spin systems, we have added examples of antiferromagnetic Ising model, XXZ model including both ferromagnetic interactions and XY phase, GHZ states and W states with random local rotations, to test the performance of our model. The results on six-qubit scenario are shown in Table I and the results on 10-, 20- and 50-qubit scenario are shown in Figure 2. Physical analysis on the performances is also added.

3. The effects of measurement noises in test data are included in both Table I for multi-qubit states and Table II for continuous-variable states. Besides average performance, we also include worst-case classical fidelity in Table II for continuous-variable states and Figure 3 for online state learning.

4. We add a figure of 2D embedding of state representations of six-qubit states as well as discussions about clustering of six-qubit state representations.

5. In “Discussion” section, we add the discussion that GQNN can be used, in an unsupervised way, to predict outcome statistics of measurements not performed yet solely from measurement data of the state to be characterized. The results of example cat states are presented in Table III.

6. We also briefly discuss about settings of hyperparameters, learning random multi-qubit states, overfitting due to unbalanced training data, and generalization performance for an informationally incomplete set of measurements in the main text, while present the detailed results in the Supplementary Notes.

REVIEWER COMMENTS

Reviewer #1 (Remarks to the Author):

In this substantially improved resubmission, the authors address all my comments in a satisfactory way.

I believe that the presence in the main text of the spin models, which are familiar to a bigger audience, has improved the presentation of the method.

The authors also explicitly discuss the current drawbacks of the method.

I also read the other referee reports as well as the replies.

Overall, I believe that the improved manuscript is now ready for publication.

Reviewer #2 (Remarks to the Author):

Thank you, the authors have adequately addressed my concerns. I recommend acceptance.

Reviewer #3 (Remarks to the Author):

I appreciate the great effort that the authors have made in expanding, clarifying, and adding more numerical experiments, and I thank them for their answers to my questions. The new version of the manuscript is a major improvement both in clarity and completeness. While I'm inclined to make a recommendation for publication, for me there is still a point from my previous report which was not entirely made clear by the authors in their response. This is regarding the functioning of the algorithm when there are multiple types of states in the training set.

I understand (as the authors further stress in their response) that, in the case of a single type of state ρ , r_i are representation vectors of ρ for each measurement i and therefore the aggregate $r = \sum_i r_i$ is unambiguously a representation vector of ρ . Then, with multiple types of states in the training set, one just repeats the procedure, "each time using the data from a different quantum state, but without taking an average over the various states under consideration." (quoting the authors' response letter).

This means to me that, during training, there will be a collection of aggregated representation vectors $\{r^k\}$ where k would be an index labelling the different types of states that the network has so far seen. The subtlety would be that r^2 is *not* a representation of the state of type 2, but what the representation network produces after seeing data from type 1 AND type 2. This may not be the same as the average of independent representation vectors of states 1 and 2. Is this correct?

If so, then the r that is finally fed as input of the generation network in the prediction phase is a single representation vector that takes into account all data seen in the training which includes different types of states. So r is a representation of the fiducial set of states. Assuming that this reasoning is correct, I have one question and one suggestion:

1) As I said in my previous report, I would be interested in seeing whether there is a difference between this final r , and the r that would result from measurement data coming directly from the averaged state of the whole training set. Particularly, in the toy example that I proposed where the training is comprised by an equal number of orthogonal states of a single qubit (a setting where data would be conflicting and I would not expect any learning algorithm to work well), I would expect this r to be not very different from one arising from the completely mixed state.

2) The suggestion is to further clarify in the manuscript what exactly is the input r of the generation network in the prediction phase when multiple types of states are considered. In the Methods section, "Network testing" subsection, there is no mention of r as input of the generation network, only measurements S in M . Yet, in several parts of the manuscript it is said that to make a prediction the

network needs an input pair (r, m') (e.g. algorithm 1's pseudocode and fig 3 in the supplementary note 1), without spelling out what r is in the case of different types of states.

We thank Referees 1 and 2 for their positive recommendations toward publication, and Referee 3 for the positive consideration and for suggesting a valuable clarification, now included in the revised version.

The Referee's comments are reproduced verbatim and in full below as indented blue font, and our point-by-point responses are in non-indented black font.

RESPONSE TO REVIEWER 3

I appreciate the great effort that the authors have made in expanding, clarifying, and adding more numerical experiments, and I thank them for their answers to my questions. The new version of the manuscript is a major improvement both in clarity and completeness.

We thank the Referee for the appreciative comments, and for the positive assessment of the revised version.

While I'm inclined to make a recommendation for publication, for me there is still a point from my previous report which was not entirely made clear by the authors in their response. This is regarding the functioning of the algorithm when there are multiple types of states in the training set.

Thank you for this valuable point. In the revised version, we have clarified it, providing an explicit discussion of how the algorithm has to be applied when multiple states are involved.

I understand (as the authors further stress in their response) that, in the case of a single type of state ρ , r_i are representation vectors of ρ for each measurement i and therefore the aggregate $r = \sum_i r_i$ is unambiguously a representation vector of ρ .

Correct.

Then, with multiple types of states in the training set, one just repeats the procedure, "each time using the data from a different quantum state, but without taking an average over the various states under consideration." (quoting the authors' response letter).

This is also correct, although the subsequent comments in your referee report indicate that there might have been an ambiguity as to what we meant by “repeating the procedure.” What we meant is that, when there are multiple states $\rho^{(j)}$, $j \in \{1, \dots, K\}$, the same procedure is used to compute a state representation vector $\mathbf{r}^{(j)}$ for every state $\rho^{(j)}$: first, GQNN produces one representation vector $\mathbf{r}_i^{(j)}$ for each measurement i , and then aggregates the representation vectors $\left(\mathbf{r}_i^{(j)}\right)_{i=1}^s$ into the average vector $\mathbf{r}^{(j)} := \frac{1}{s} \sum_i \mathbf{r}_i^{(j)}$. As a consequence, the vector $\mathbf{r}^{(j)}$ is unambiguously a representation of the state $\rho^{(j)}$, just as \mathbf{r} was a representation of ρ in the single-state scenario.

In the revised version, we described explicitly how the algorithm is applied when GQNN is used to characterize multiple states $\rho^{(j)}$, $j \in \{1, \dots, K\}$.

This means to me that, during training, there will be a collection of aggregated representation vectors $\{r^k\}$ where k would be an index labelling the different types of states that the network has so far seen.

This seems to be the source of the misunderstanding. The index k (or j in the above discussion) labels a specific quantum state $\rho^{(k)}$, not all the quantum states that the network has seen until the k -th time step. GQNN produces a representation vector $\mathbf{r}^{(k)}$ for every state (i.e. for every density matrix $\rho^{(k)}$) under consideration. State representation vectors corresponding to different density matrices are never aggregated into a single vector.

The subtlety would be that r^2 is *not* a representation of the state of type 2, but what the representation network produces after seeing data from type 1 AND type 2.

This point should be clear now: the vector $\mathbf{r}^{(j)}$ is just a representation of the state $\rho^{(j)}$, not an aggregate representation of all the states $\rho^{(k)}$ with $k \leq j$.

This may not be the same as the average of independent representation vectors of states 1 and 2. Is this correct?

Yes, definitely $\mathbf{r}^{(2)}$ is not an average of representation vectors for states 1 and 2. But the point is not whether the vectors are independent or not: rather, it is that GQNN never computes an average of state representation vectors corresponding to different states.

If so, then the r that is finally fed as input of the generation network in the prediction phase is a single representation vector that takes into account all data seen in the training which includes different types of states. So r is a representation of the fiducial set of states.

No, this is not the case; thank you making us aware of the potential confusion that could lead to this incorrect conclusion. The purpose of the training is *not* to produce a representation vector of the fiducial set of states, but to optimize the value of the parameters ξ and η . The parameter ξ is then used to compute the representation vectors of the states appearing in the prediction phase. For the j -th state $\rho^{(j)}$, the state representation is $\mathbf{r}^{(j)} := \frac{1}{s} \sum_{i=1}^s f_{\xi}(\mathbf{m}_i^{(j)}, \mathbf{p}_i^{(j)})$, where $\mathbf{p}_i^{(j)}$ is the measurement statistics obtained by measuring the state $\rho^{(j)}$ with the measurement $\mathbf{m}_i^{(j)}$, the i -th measurement of a randomly chosen set of s measurements $\mathcal{S}^{(j)}$ (note that, in general, the set of performed measurements $\mathcal{S}^{(j)}$ may depend on j).

In the revised version, this point is clarified in the Results and Methods sections.

Assuming that this reasoning is correct, I have one question and one suggestion:

1) As I said in my previous report, I would be interested in seeing whether there is a difference between this final r , and the r that would result from measurement data coming directly from the averaged state of the whole training set. Particularly, in the toy example that I proposed where the training is comprised by an equal number of orthogonal states of a single qubit (a setting where data would be conflicting and I would not expect any learning algorithm to work well), I would expect this r to be not very different from one arising from the completely mixed state.

As it should be clear from the above discussion, in our model there does not exist a final \mathbf{r} that aggregates all the state representations of states in the training set.

2) The suggestion is to further clarify in the manuscript what exactly is the input r of the generation network in the prediction phase when multiple types of states are considered. In the Methods section, “Network testing” subsection, there is no mention of r as input of the generation network, only measurements $S \in M$. Yet, in several parts of the manuscript it is said that to make a prediction the

network needs an input pair (r, m') (e.g. algorithm 1's pseudocode and fig 3 in the supplementary note 1), without spelling out what r is in the case of different types of states.

Thank you very much for this valuable suggestion, much appreciated especially in the light of the above comments. In the revised version, we included the explicit discussion of multiple state case in the “Network testing” subsection and in Supplementary Note 1.

REVIEWERS' COMMENTS

Reviewer #3 (Remarks to the Author):

The authors have satisfactorily clarified my doubts and again improved the manuscript accordingly. I recommend acceptance.